# Superresolution architecture of cornerstone focal adhesions in human pluripotent stem cells

Aki Stubb[1,6], Camilo Guzmán[1,4,6], Elisa Närvä[1], Jesse Aaron [2], Teng-Leong Chew[2], Markku Saari[1], Mitro Miihkinen[1], Guillaume Jacquemet [1,5] & Johanna Ivaska [1,3]*

While it is clear that key transcriptional programmes are important for maintaining pluripotency, the requirement for cell adhesion to the extracellular matrix remains poorly defined. Human pluripotent stem cells (hPSCs) form colonies encircled by an actin ring and large stable cornerstone focal adhesions (FA). Using superresolution two-colour interferometric photo-activated localisation microscopy, we examine the three-dimensional architecture of cornerstone adhesions and report vertical lamination of FA proteins with three main structural features distinct from previously studied focal adhesions: 1) integrin β5 and talin are present at high density, at the edges of cornerstone FA, adjacent to a vertical kank-rich protein wall, 2) vinculin localises higher than previously reported, displaying a head-above-tail orientation, and 3) surprisingly, actin and α-actinin are present in two discrete z-layers. Finally, we report that depletion of kanks diminishes FA patterning, and actin organisation within the colony, indicating a role for kanks in hPSC colony architecture.

[1] Turku Bioscience Centre, University of Turku and Åbo Akademi University, FI-20520 Turku, Finland. [2] Advanced Imaging Center, HHMI Janelia Research Campus, Ashburn, VI 20147, USA. [3] Department of Biochemistry, University of Turku, FIN-20520 Turku, Finland. [4]Present address: Nanophotonics and Bioimaging Facility, INL—International Iberian Nanotechnology Laboratory, Av. Mestre José Veiga s/n, 4715-330 Braga, Portugal. [5]Present address: Faculty of Science and Engineering, Cell Biology, Åbo Akademi University, 20520 Turku, Finland. [6]These authors contributed equally: Aki Stubb, Camilo Guzmán. *email: joivaska@utu.fi

Human pluripotent stem cells (hPSC) have the capacity to self-replicate and differentiate into virtually any cell type in the human body. This combined with the seminal finding that hPSC can be generated by artificial reprogramming of somatic cells[1,2], underscores the immense therapeutic potential of these cells. The delineation of transcriptional programmes that control pluripotency, as well as the development of protocols to induce specific differentiation programmes are at the forefront of research to harness the power of hPSC[3]. Towards this end, a key consideration for the safe clinical use of hPSC, is the requirement for animal serum-free and feeder-layer-free culture conditions that support pluripotency. Important discoveries, made in the past decade, demonstrate that, in vitro, hPSC adhesion to specific extracellular matrix (ECM) molecules is fundamental for maintaining cells in a pluripotent state[4–6], thus highlighting an important role for the ECM, as well as ECM receptors, for stemness.

Cells interact with the surrounding ECM via transmembrane adhesion receptors, such as integrins, which provide a physical link between the ECM and the actin cytoskeleton[7]. Integrin engagement leads to the assembly of large signalling platforms termed focal adhesions (FA), which modulate most cellular functions[8]. FA are highly dynamic and complex structures composed of hundreds of proteins[9,10] that tune cellular responses, but their overall architecture can be studied using a handful of key adaptors. Indeed, advances in superresolution microscopy have enabled the delineation of talin-vinculin function in vivo within different drosophila tissue[11] to determine the molecular composition of specific integrin adhesion complexes[12], as well as to unravel the nanoscale architecture of FA. Previous studies have demonstrated highly organised vertical stratification of FA[13–15] using a technique that combines interferometry and photo-activated localisation microscopy (iPALM) to provide sub-20 nm, three-dimensional (3D) isotropic localisation of tagged proteins[16]. FA proteins localise to three general functional layers: a membrane proximal integrin signalling layer (composed of integrins, paxillin, and focal adhesion kinase), an intermediate force transduction layer (composed of talin and vinculin) and an actin regulatory layer (composed of both actin and actin-regulatory elements)[13–15]. Interestingly, vinculin a binding partner for many FA proteins and a key stabiliser of talin unfolding has been shown to display a wide distribution across all three layers with increasing upward positioning in mature FA[15].

Even though integrins are key requirements for maintaining pluripotency in vitro[4,5,17], and adhesion signalling through focal adhesion kinase is implicated in pluripotent stem cell maintenance[18,19], hPSC FA composition and organisation remain poorly defined. Previously, we described that the hallmark sharp hPSC colony edge morphology is regulated by unusually large FA, termed cornerstone FA, connected by an actin fence defined by strong contractile ventral stress fibres[19]. Furthermore, these colony edge adhesions exert strong traction forces on the underlying ECM, which regulates colony morphology and the orientation of cell division[19].

Here, we have used nano-grated patterns to investigate the link between FA morphology and pluripotency and a combination of high-resolution and superresolution microscopy techniques including, iPALM, structured illumination microscopy and Airyscan to determine the 3D architecture of cornerstone FA present at the periphery of hPSC colonies. We find that physical restriction of FA size, and orientation, in hPSC colonies partially compromises SOX2 and promotes SSEA-1 expression, suggesting a role for cornerstone FA architecture in supporting pluripotency. In addition, we find that the nanoscale localisation of several of the key FA components—integrins αV and β5, paxillin, vinculin, talin, actin and α-actinin—previously mapped in classical FA[13], is markedly distinct in hPSC FA. Furthermore, we include the FA scaffold partners kank1 and kank2 in our imaging, and describe a role for kanks in regulating hPSC adhesion and maintaining FA architecture in hPSCs.

## Results

**hPSC cornerstone FA contribute to pluripotency maintenance.** hPSC colonies plated on vitronectin (VTN), an ECM ligand which supports pluripotency[4], assemble into tightly packed colonies with well-defined edges[20]. Structurally, these colonies are encircled by paxillin-positive cornerstone FA that are themselves connected by prominent actin bundles termed actin fence (Fig. 1a, b[19]). Importantly, cornerstone FA or actin fences are not a general feature of colony-forming cells as they were not observed in normal epithelial cells (Supplementary Fig. 1a). Using structured illumination microscopy imaging of hPSC colonies we observed that cornerstone FA are composed of multiple linear units clustered together (Fig. 1b). To further characterise these cornerstone FA, we performed live-cell imaging of hPSC colonies, where paxillin was endogenously tagged (Fig. 1c, Supplementary Movie 1). Detailed analyses of paxillin dynamics revealed that cornerstone FA at the edge are relatively few in number but cover a large proportion of the surface at the colony edges (Fig. 1c, d). In addition, cornerstone adhesions are remarkably stable compared to FA formed at the centre of hPSC colonies (Fig. 1e, f, Supplementary Movie 1).

The unique properties of cornerstone FA (size, orientation, and stability) led us to speculate that these structures regulate hPSC colony morphology and could contribute to pluripotency maintenance. To constrain the orientation and the maximal size of FA, hPSC colonies were plated on VTN-coated nano-grated patterns of alternating 800 nm wide matrix ridges and 600 nm deep grooves (Fig. 2a–c). These nano-grids were sufficient to constrict adhesion size, reducing the characteristic area of individual FA from $4.2\,\mu m^2$ (on uniform surfaces) to $1.8\,\mu m^2$. Remarkably, on nano-grids (Fig. 2b), the characteristic rounded morphology, actin fence, and cornerstone FA of hPSC colonies were also distorted (Fig. 2a, b). Plating hPSC on nano-grids did not decrease E-cadherin protein levels or affect their ability to form E-cadherin junctions (Supplementary Fig. 1b–d), suggesting that cell–cell adhesions are not obviously disrupted. However, it remains possible that the nano-grids affect other structural parameters than the cornerstone FA and these may also contribute to the observed cellular responses. To allow for spontaneous differentiation, hPSC were maintained in E6 medium lacking pluripotency supporting growth factors bFGF and TGFβ. Under these conditions, exposure to VTN-coated nano-grids accelerated, to some extent, spontaneous differentiation as observed by a decrease in the pluripotency factor Oct4 (day 1), Sox2 (days 3 and 6) and, conversely, an increase in the level of the differentiation marker SSEA-1 (Fig. 2d–f). In contrast, cells maintained on uniform VTN-coated surfaces remained Sox2 positive and SSEA-1 low (Fig. 2d–f). The pace of the nano-grid-induced effect was not as rapid as the BMP-4-induced response (used as a positive control for differentiation) (Fig. 2d–f). However, the nano-grid-induced differentiation appeared to be irreversible as cells grown on nano-grids for 3 days were not able to recover high expression levels of pluripotency factors when replated on VTN-coated surfaces in normal culture condition (E8 medium, Supplementary Fig. 1e–g). These data suggest that physical constriction of FA size and orientation in hPSC colonies could support spontaneous differentiation. However, more work would be needed for definitive proof of a direct causal relationship between FA size and orientation and maintenance of pluripotency.

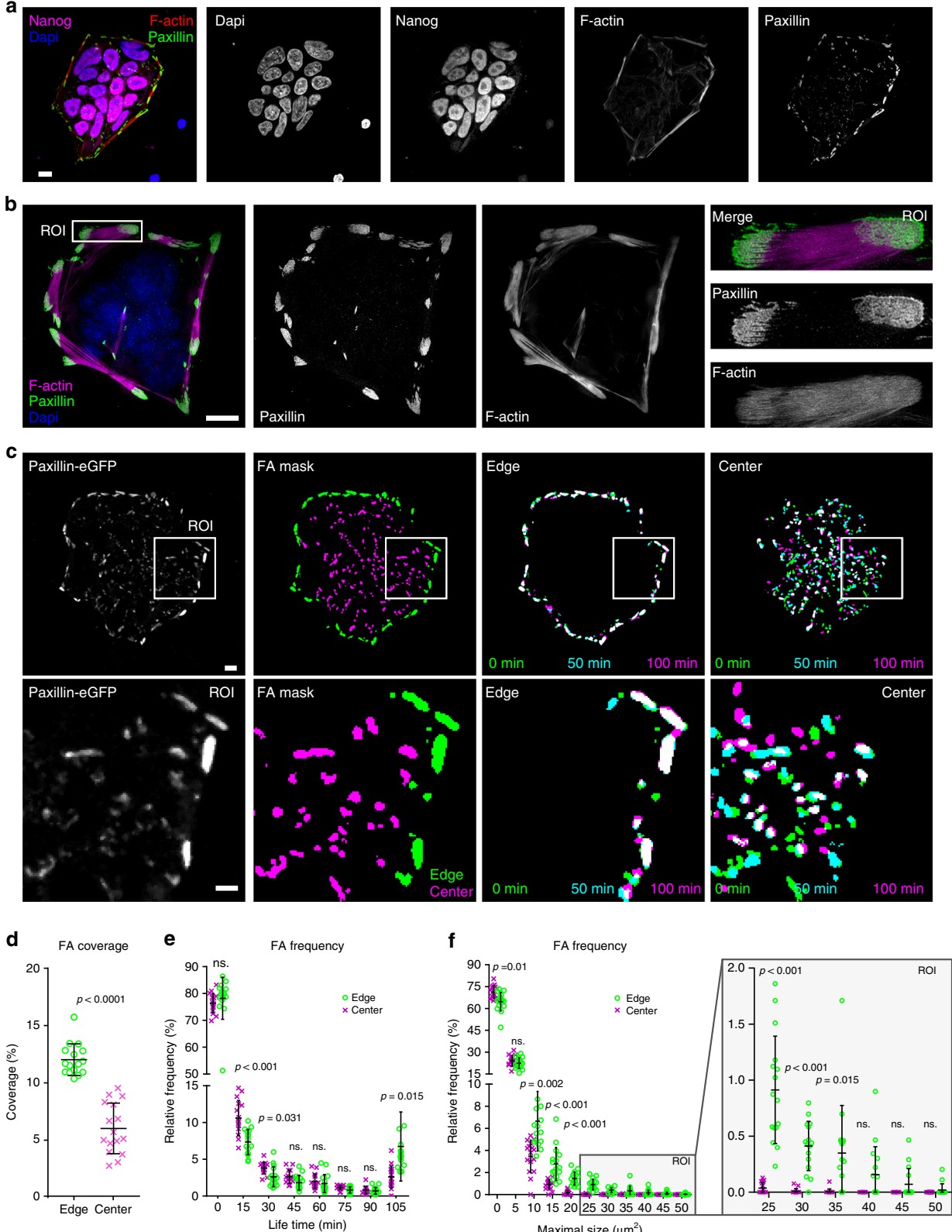

These observations are in line with previous work reporting that hPSCs differentiate faster when stimulated with the morphogen BMP-4 when cultured on a 650 nm nano-grated pattern compared to a uniform substrate[21]. Together, these data indicate a link between cornerstone FA architecture and pluripotency maintenance, and prompted us to investigate the nanoscale architecture of these adhesions in more detail.

**iPALM imaging of hPSC cornerstone FA.** To deconstruct the ultrastructure of hPSC cornerstone FA, we used iPALM, a strategy that enables sub-20 nm 3D localisation of molecules[16]. Previous studies employing iPALM for analysis of osteosarcoma (U2OS) cell adhesions, uncovered FA architecture as a well-organised hierarchy of at least three distinct, vertically separated, functional layers (the integrin signalling layer, the force

**Fig. 1** Characterisation of cornerstone adhesions. **a** Spinning disk images of hPSC plated on vitronectin (VTN) and stained for F-actin, paxillin, DAPI, and the pluripotency marker Nanog. Scale bar 10 μm. **b** Structured illumination microscopy images of hPSC plated on VTN and stained for filamentous actin (F-actin) and paxillin. Scale bar 10 μm. The white square highlights a region of interest (ROI), which is magnified. **c–f** Live-cell imaging of endogenously tagged paxillin in hPSC. Images were acquired every minute using a spinning disk microscope. Colonies were recorded for at least 105 min ($n = 16$ independent colonies from three biologically independent experiments). A representative video is available as supplementary information (Supplementary Movie 1). **c** A representative image is displayed (ROI highlighted by white square and magnified; size, 30 μm) in addition to a mask image highlighting paxillin-positive adhesions (colony edge FA, green; colony centre FA, magenta) and merged images depicting paxillin-positive adhesion lifetime (selected time points 0 min, green; 50 min, cyan and 100 min, magenta) within one hPSC colony (white represents very stable adhesions). **d** The percentage area covered by paxillin-positive adhesions at the edge or at the centre of hPSC colonies was measured over the duration of the movies. Statistical significance was determined by Student's $t$-test (two-tailed, unpaired). **e, f** The frequency distributions of paxillin-positive adhesion lifetime (**e**) and maximal size (**f**) (colony edge FA, green circle; colony centre FA, magenta cross) within hPSC colonies are displayed. Each data point represents the frequency distributions in one hPSC colony. Statistics: multiple $t$-tests with correction using Holm–Sidak method. Error bars depict standard deviation. Source data are provided as a Source Data file

transduction layer, and the actin regulatory layer)[13]. To assess the organisation of hPSC cornerstone FA, representative proteins were chosen from each of these layers for imaging, including VTN-binding integrins αV and β5, and paxillin (integrin signalling layer), talin-1 and vinculin (core components of the force transduction layer), and α-actinin-1 and actin (actin-regulatory layer)[13]. In addition, both kank1 and kank2, two recently described FA-associated proteins, were included due to their unique localisation at the outer rim of FA[22,23]. Of note, the long-isoform of kank1[23] is highly expressed in hPSC (Supplementary Fig. 2), and was therefore selected over other isoforms for imaging. Importantly, we could confirm both the endogenous expression of the selected panel of proteins in hPSC (Supplementary Fig. 2) and localisation to cornerstone FA and/or the actin cytoskeleton (Supplementary Fig. 3).

Imaging of endogenous molecules by iPALM is however challenging, due to the requirement for photo-switchable labels as well as high-density labelling[16]. Therefore, and as previously described by others in different cell types[13–15], for the purpose of gaining superresolution detail into hPSC FA, proteins of interest, tagged with the photoactivatable fluorescent protein (PA-FP) Eos, were transiently expressed in hPSC. In addition, to image kank1 and kank2, Eos-tagged constructs were generated (see the "Methods" section for details). We then validated that all Eos-tagged proteins exhibited similar localisation to the endogenous proteins and that the transient expression of these constructs did not affect colony morphology or levels of the pluripotency marker Oct4 (Supplementary Fig. 4).

**Talin-1 and β5 integrin accumulate at the FA periphery.** Using iPALM, we first determined the lateral distribution of proteins localising to the various FA functional layers (detailed values for iPALM data are included in Supplementary Table 1). All FA proteins, with the exception of β5 integrin and talin-1, displayed homogeneous $x–y$ distribution and density within cornerstone FA. On the other hand, β5 integrin and talin-1 revealed an obvious ring-like distribution, with higher protein density at the edges of cornerstone adhesions (Fig. 3a–c). The integrin α subunit partner for β5, αV, was, however, homogenously distributed, likely reflecting the known interaction of αV with multiple other β-integrin subunits[24].

The accumulation of Eos-tagged β5 integrin and talin to the rim of the large cornerstone FA was recapitulated with staining of the endogenous proteins (Fig. 3d, Supplementary Figs. 5–7). In this case, αVβ5 integrin (detected either with an antibody recognising the β5 subunit or the αVβ5 heterodimer) was enriched in cornerstone FA edges (Fig. 3d, Supplementary Figs. 6 and 7), while active or total β1 integrin and paxillin were distributed uniformly throughout the interior of the adhesion (Fig. 3d, Supplementary Figs. 5–7). β3 integrin was only weakly

detected in hPSC and did not appear to accumulate to cornerstone FA (Supplementary Fig. 8). Interestingly the ring-like distribution of talin could not be observed in normal epithelial cells (Supplementary Fig. 5e, f) and αVβ5 integrin and β1 integrin predominantly segregated to distinct FA rather than being segregated into different subdomains within the same FA (Supplementary Fig. 6e). Taken together, in hPSC cornerstone FA, the integrin signalling layer appears to be horizontally segregated into sub-regions of different ECM-binding integrins.

**Talin is fully extended within cornerstone FA.** We next used iPALM to determine the vertical $z$ positioning of the chosen adhesion proteins (distance measured from the coverslip) (Fig. 3e, detailed values for iPALM data are included in Supplementary Table 1). We found that the components of the integrin signalling layer (integrins β5 and αV, and paxillin) have a similar vertical distribution in hPSC cornerstone FA to those reported for U2OS FA with $z$-positions indicating a close relationship with the cell membrane (Fig. 3f). However, talin-1 and vinculin, components of the force transduction layer, exhibited a higher $z$-position than previously observed[13–15] (Fig. 4a–c).

Talin can adopt multiple conformations in the cell, ranging from a head-to-tail auto-inhibited conformation to a fully extended, mechanically stretched integrin-bound and actin-bound conformation[25]. To evaluate the 3D FA distribution of talin as well as its vertical extension, we imaged both N-terminally (talin-1-N) and C-terminally (talin-1-C) tagged talin-1[14,26]. Overall, the $z$-position values for talin-1-N ($Z_{centre} = 72.8 \pm 12$ nm, ± represent the standard deviation) and for talin-1-C ($Z_{centre} = 103.5 \pm 20.5$ nm) were higher than those reported for human umbilical vein endothelial cells (HUVECs)[14]. However, the distance between the two ends of talin was comparable to that previously measured and considered to represent the extended talin conformation[14,15,26]. Moreover, calculations of the inclination angle ($\theta \approx 17°$, based on the vertical separation of the two talin ends and on the full-length of the protein) were supportive of an extended talin molecule, in agreement with previous work[14]. Thus, our results indicate that talin is fully extended throughout the cornerstone FA, clustering at a higher density at the FA edge (Figs. 3c and 4b).

**Vinculin displays a head-above-tail orientation in FA.** In classical FA, vinculin displays a wide vertical distribution and has a preferred orientation with the tail above the head[15]. iPALM analyses of N-terminally (vinculin-N) and C-terminally (vinculin-C) tagged vinculin constructs (Fig. 4c, Supplementary Fig. 9a) revealed that the vinculin vertical positioning in hPSC cornerstone adhesions is significantly higher than in classical FA[13,15]. Increased vinculin $z$ positioning has been linked to vinculin

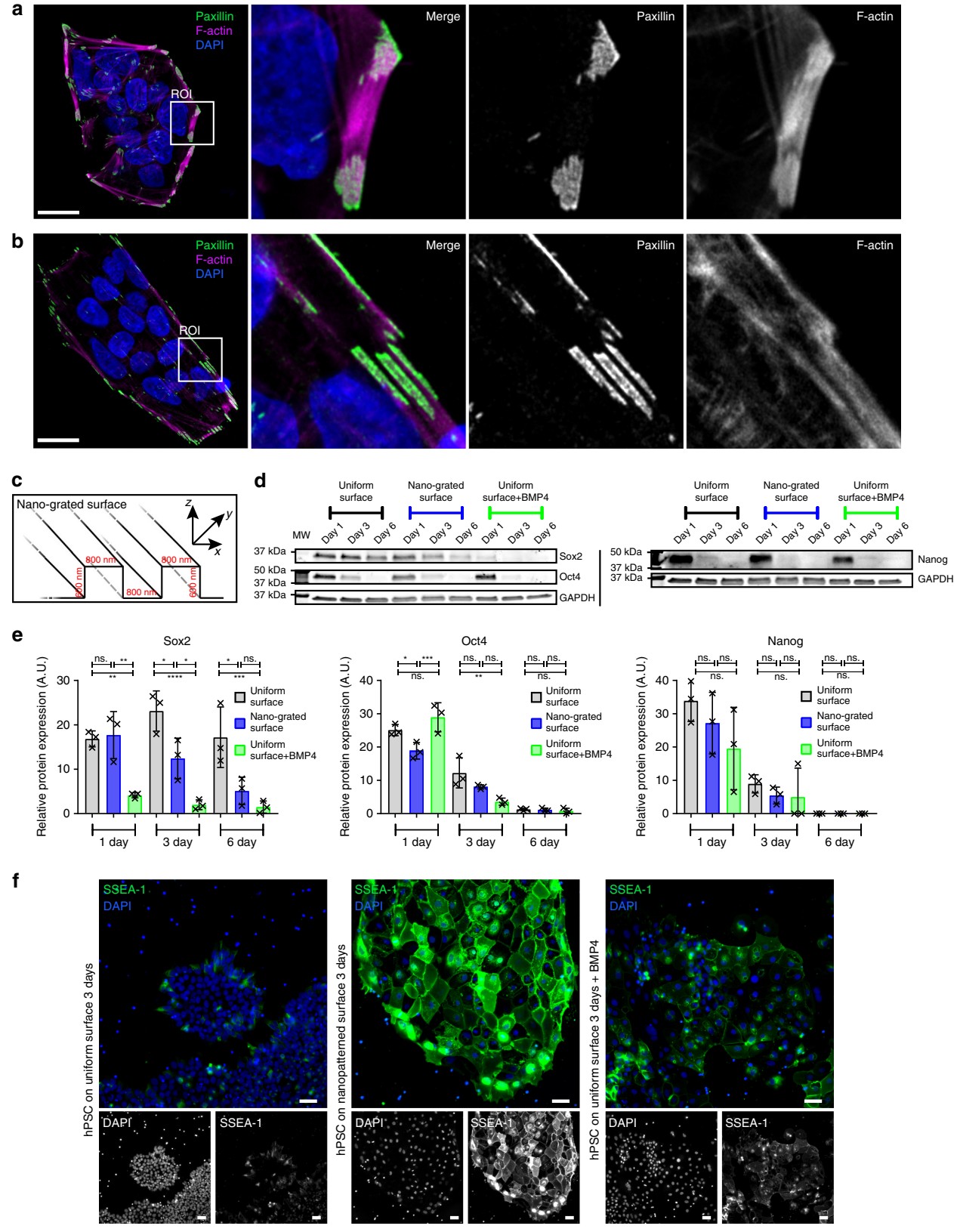

activation and FA maturation[15], suggesting that hPSC cornerstone adhesions may contain active vinculin. Very unexpectedly, vinculin was oriented head above the tail in hPSC cornerstone adhesions (vinculin-N, $Z_{centre} = 100.4 \pm 14.4$ nm; vinculin-C, $Z_{centre} = 76.4 \pm 16.3$ nm) (Fig. 4c). To verify this unexpected

observation, we set-up a two-colour iPALM strategy, where Eos-tagged vinculin-C was imaged together with endogenous paxillin (detected with Alexa-647-labelled antibody) at high resolution in the same cells. Using this approach we were able to validate that the vinculin tail is indeed in close relationship with the integrin

**Fig. 2** Constraining cornerstone adhesions accelerates spontaneous differentiation. **a**, **b** hPSC plated for 24 h on a VTN-coated uniform surface (**a**) or on a VTN-coated nano-grated surface (nano-grids) (**b**) in normal (E8 medium) culture conditions, were stained for F-actin, paxillin, and DAPI. Images were acquired using an airyscan confocal microscope. Scale bars 20 μm. A region of interest (ROI) is magnified in each image. **c** Cartoon highlighting the topography of the nano-grated surface. **d**, **e** Representative western blot (**d**) and quantification (**e**) of Sox2, Oct4, and Nanog levels in hPSC plated for 1, 3, or 6 days either on VTN-coated uniform surfaces (black/grey), in the presence or absence of BMP4 (green), or on VTN-coated nano-grids (blue) in basal E6 medium (n = 3 biologically independent experiments; two times using hiPSC line HEL 24.3 and 1 time using hiPSC line HEL 11.4). Statistics: one-way ANOVA with multiple comparisons complemented with Bonferonni's post-hoc test. **f** hPSC plated for 3 days either on VTN-coated uniform surfaces, in the presence of basal E6 medium and/or BMP4, or on VTN-coated nano-grids, in the presence of basal E6 medium, were stained for SSEA-1 and DAPI. Images were acquired on a spinning disk confocal microscope using a ×20 objective (n = 3 biologically independent experiments). Scale bars 50 μm. Source data are provided as a Source Data file. Error bars depict standard deviation

signalling layer (Fig. 4d). This is markedly distinct from the previously described tail above head vinculin orientation in somatic cells and indicates that vinculin is upside down in hPSC cornerstone FA. This very peculiar orientation of vinculin was also recently reported in FA of mouse embryonic stem cells plated on laminin, but not gelatin or fibronectin[27]. A head-above-tail vinculin conformation could be indicative of an extreme activation state, potentially triggered by the high forces exerted in cornerstone FA[19]. Indeed, vinculin has been reported to dissociate from talin in cases where forces borne by an individual protein exceed 25 pN[28]. In such cases, we speculate that the vinculin head could bind a different partner that localises higher up in cornerstone FA (such as α-actinin-1). The vertical position of the first α-actinin-1 ($Z_{centre} = 125.5 \pm 15.1$ nm) layer partially overlaps with the distribution of vinculin-N ($Z_{centre} = 100.4 \pm 14.4$ nm). Alternatively, the vinculin tail could engage components close to the plasma membrane such as paxillin[29]. Indeed, the vertical position of paxillin ($Z_{centre} = 55.7 \pm 8.5$) partially overlaps with the distribution of vinculin-C ($Z_{centre} = 76.4 \pm 16.3$ nm).

In addition to an inverted orientation, 3D rendering of the acquired localisations for vinculin revealed that its distribution is not homogeneously flat within cornerstone FA but rather that its distribution forms a solid paraboloid or cup-like shape (Fig. 4e). This cup-like shape appeared to be specific for vinculin, as it was not observed for paxillin (Fig. 4e), and indicates that membrane distal vinculin molecules occupy a larger lateral area than the membrane proximal vinculin (Fig. 4f). The spatially controlled intra-FA z-distribution of vinculin (Fig. 4e), as well as the ring-shaped distribution of talin and β5 integrin (Fig. 3), suggest that the edge of the cornerstone FA is architecturally unique from its centre.

The unexpected 3D orientation of vinculin in hPSC cornerstone FA prompted us to investigate the role of vinculin in hPSC colony morphology and pluripotency maintenance. We used lentiviral shRNA to silence 95% of endogenous vinculin (Supplementary Fig. 9b). Imaging of control and shvinculin hPSC colonies on VTN showed that, surprisingly, loss of vinculin does not disrupt colony formation, the actin fence or cornerstone FA (Supplementary Fig. 9c and d). Vinculin silencing appeared to only modestly increase the size of the small FA found at the centre of hPSC colonies. In addition, silencing of vinculin did not affect Oct4 expression levels (Supplementary Fig. 9b) indicating that vinculin is not required for the maintenance of pluripotency under these conditions.

**Cornerstone adhesions have two actin layers**. Cornerstone FA are connected by prominent ventral actin bundles (Fig. 1a, b), and therefore actin is found both inside and outside FA. To determine accurately the vertical positioning of actin in hPSC cornerstone adhesions, endogenous paxillin was used as a reference marker and two-colour iPALM was performed (Fig. 5a, b). Unlike the proteins in the integrin signalling and the force transduction

layers, actin distribution within many cornerstone FA was not Gaussian but could instead be described as the sum of two Gaussian distributions (Fig. 5c). This indicates that, much to our surprise, and in contrast to conventional FA nanoscale architecture[13], actin can form two coexisting vertical layers separated by a ~50 nm gap on top of the cornerstone FA (Fig. 5b, c).

This axial actin localisation pattern was further confirmed by the distribution of α-actinin-1, a protein typically overlapping with actin[30]. α-actinin-1 could also present the same two-peak distribution than actin albeit with a slight shift in z positioning (Fig. 5d, e). Importantly, the separation between the two actin peaks and the two α-actinin-1 peaks was identical suggesting that each actin layer has a corresponding α-actinin-1 layer. The vertical position of the first actin ($Z_{centre} = 99.6 \pm 14.4$ nm) and α-actinin-1 ($Z_{centre} = 125.5 \pm 15.1$ nm) peaks matches the z position reported for these proteins in U2OS cells[13]. In contrast, the second peak for actin ($Z_{centre} = 155.4 \pm 14.2$ nm) and α-actinin-1 ($Z_{centre} = 174.8 \pm 17.1$ nm) represents a higher-localising actin population (Fig. 5f), not described before. In the future, it would be interesting to determine the mechanisms regulating this vertical separation of the actin cytoskeleton and the purpose of the two layers and to identify the proteins positioned within the ~50 nm gap.

**Nanoscale localisation of kank1 and kank2**. While cornerstone FA display 3D stratification of integrin signalling, force transduction, and actin regulatory layers, albeit with substantial deviations from those described previously[13–15], there is a clear distinction in protein distribution profiles between the FA edge and centre. Therefore, we proceeded to search for possible scaffold proteins that could uniquely mediate the segregation of the cornerstone FA. Evolutionary conserved scaffold proteins kank1 and kank2 localise to the border of FA and form links with talin and the actin cytoskeleton[22,23].

We performed two-colour iPALM of Eos-tagged kank1 and kank2 together with endogenous paxillin to enable precise determination of kank 3D localisation with respect to FA (Fig. 6a–d). Kank1 localised predominantly at the border of paxillin-positive FA with individual molecules detected inside adhesions (Fig. 6a, b). This localisation was validated with endogenous staining and high-resolution imaging of kank1 (Supplementary Fig. 3). In contrast, kank2 was always present at the outer rim and/or outside paxillin-positive FA in hPSC (Fig. 6c). In addition, kank2 only accumulated near cornerstone FA, always towards their distal end, and occasionally following the direction of ventral stress fibres (Fig. 6c). Vertically, both kank1 and kank2 localisation varied depending on their proximity to FA. At the outer rim of FA, both kank1 and kank2 displayed a higher z position than paxillin (60 nm above paxillin), corresponding to the height of the force transduction layer (Fig. 6a, c, d, Supplementary Movie 2). Interestingly, in paxillin-negative structures, both kank1 and kank2 localised at a lower vertical position indicating a close proximity to the plasma

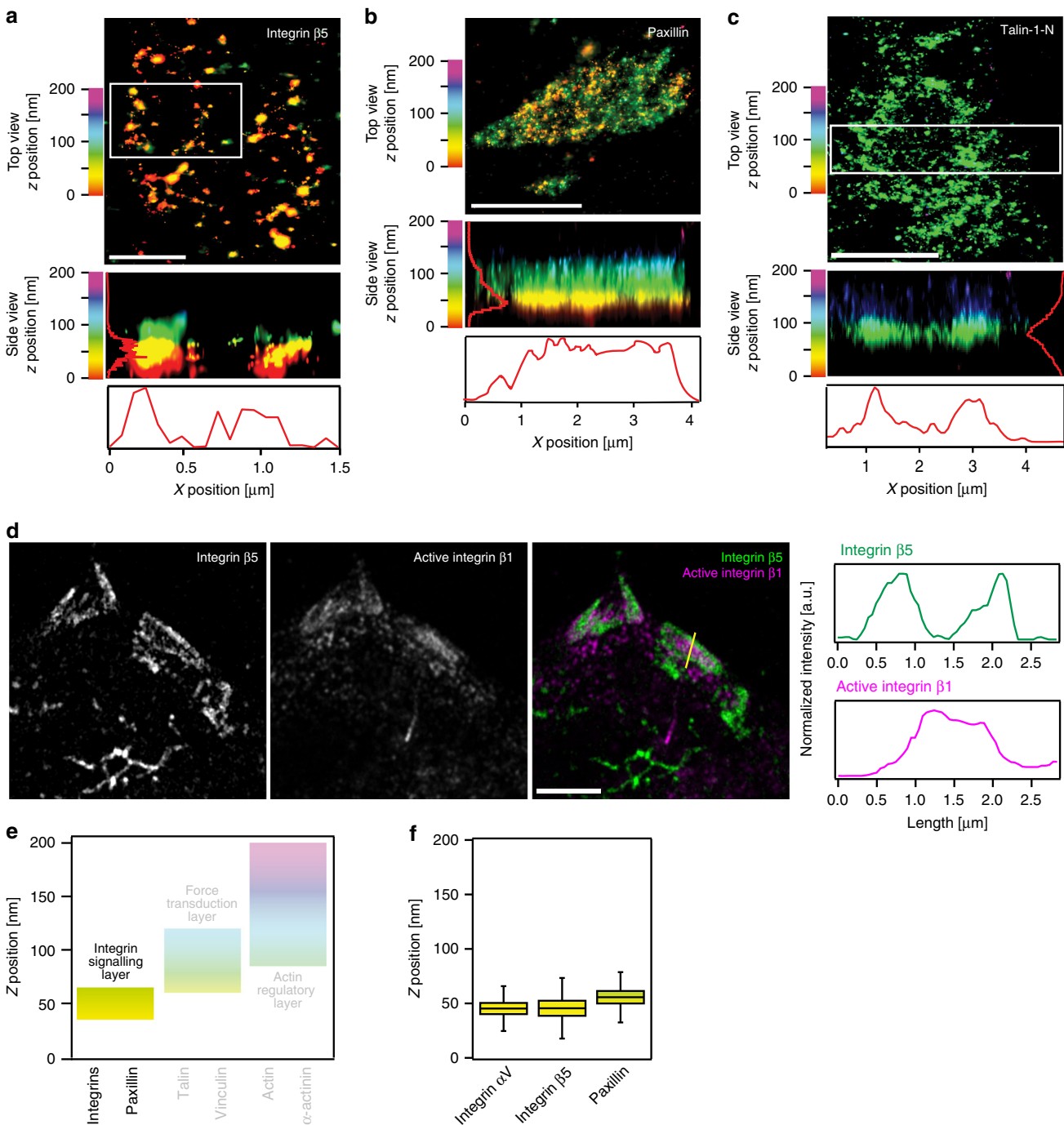

**Fig. 3** Lateral and vertical segregation of proteins within cornerstone FA. **a–c** Interferometric photo-activated localisation microscopy (iPALM) images of Eos-tagged integrin β5 (**a**), paxillin (**b**), and talin-1-N (N-terminally tagged talin-1) (**c**) in cornerstone FA. Individual cornerstone FA are displayed. Both top-view (*xy*) and side-view (*xz*) images are displayed. White boxes indicate the area used to generate the side-view images (in the absence of a white box, the entire image was used). For all images, individual localisations are colour-coded as a function of their *z*-positioning (distance from the coverslip). In addition, normalised line intensity profiles of the region of interest, in both *z*- and *x*-axis, are shown in red lines. **d** hPSC plated for 24 h on VTN were stained for endogenous integrin β5 and active integrin β1 (mAb 12G10) and imaged using an Airyscan confocal microscope. The yellow line highlights the area used to measure the intensity profiles displayed on the side. Scale bars (**a**) 1 μm, (**b**, **c**) 2 μm, (**d**) 5 μm. **e** Schematic representation of the different FA layers previously identified using iPALM in U2OS cells[13], colour bars highlight the *z* range for each of the three layers and the integrin signalling layer is emphasised over other layers. **f** iPALM analysis of the *z* position (distance from the coverslip, $Z_{centre}$) of some of the integrin signalling layer components, Eos-tagged integrins αV (*n* = 17 biologically independent ROI. Given that cornerstone FAs are clusters of FAs, one ROI may contain multiple FAs. For details on the number of localisations, and HPSC colonies see supplementary Table 1) and β5 (*n* = 22 biologically ROI), and Eos-tagged paxillin (*n* = 23 biologically independent ROI), in hPSC cornerstone adhesions. Boxes display the median, plus the 1st and 3rd quartiles (IQR: 25th– 75th percentile). Whiskers correspond to the median ± 1.5 × IQR. Source data are provided as a Source Data file

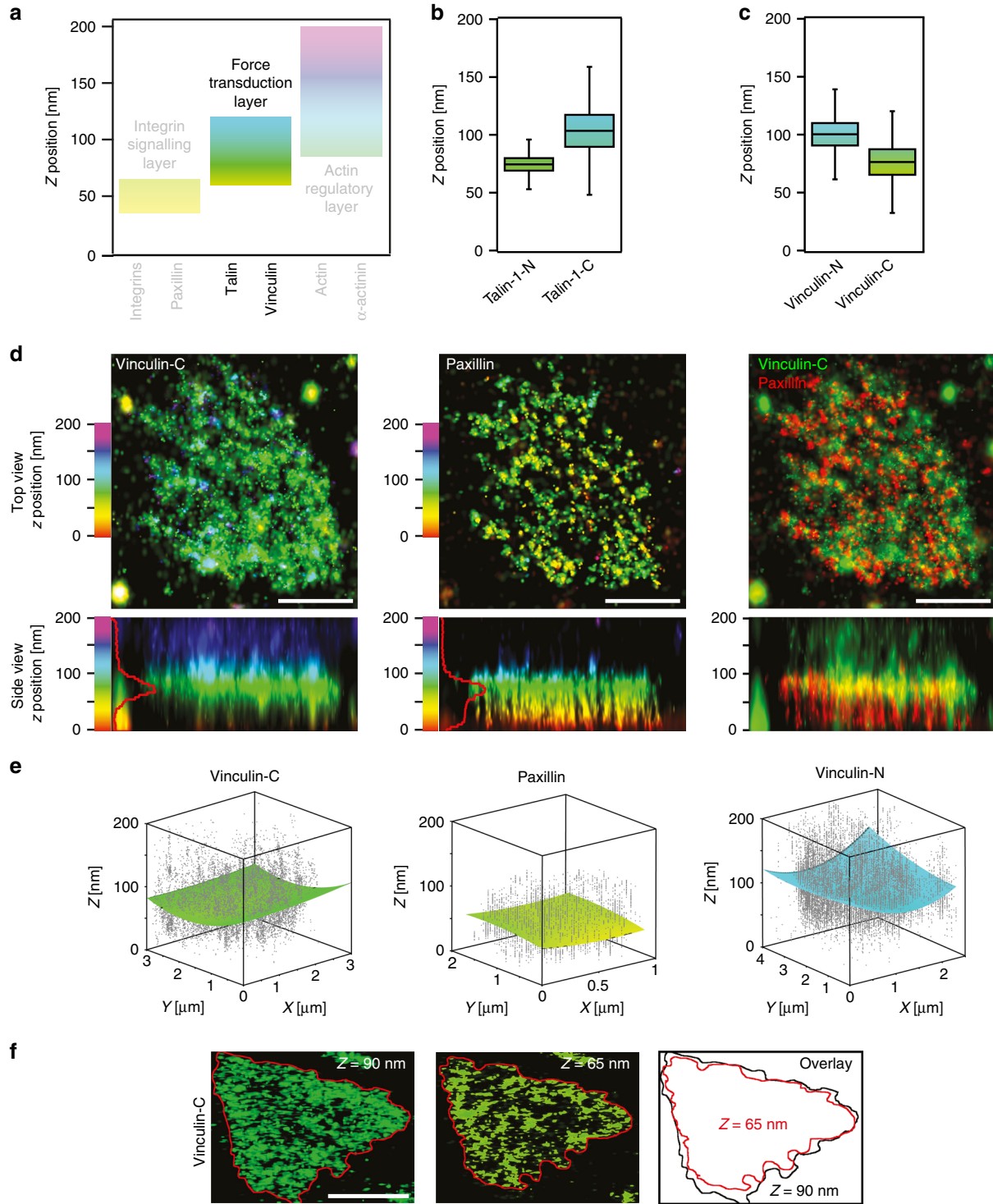

membrane (Fig. 6b–d). The z positions obtained for kank1 and kank2 adjacent to FA were $Z_{centre} = 116.1 \pm 12.1$ nm and $Z_{centre} = 100.8 \pm 13.4$ nm and for FA-distal kank1 and kank2, $Z_{centre} = 63.4 \pm 13.6$ nm and $Z_{centre} = 74.2 \pm 17.7$ nm, respectively (Fig. 6d).

Kank proteins modulate adhesion size and properties in somatic cells by both regulating integrin function and microtubule targeting[31]. As both kank1 and kank2 localise to the outer rim of hPSC cornerstone adhesions, we wanted to investigate the contribution of kank1 and kank2 to cornerstone adhesion properties. Unexpectedly, downregulation of kank1 and kank2 expression using siRNA led to smaller cornerstone FA at the colony edge, and, conversely, to larger FA at the centre of the

colonies (Fig. 6e, f, Supplementary Fig. 10). Thus, kank1 and kank2 are important regulators for intra-colony distinction of colony centre and edge cells and very interestingly differentially regulate the dynamics of the large cornerstone FA and the smaller adhesions in the colony centre.

## Discussion
An important prerequisite for maintaining hPSC pluripotency in vitro is cell adhesion to specific ECM molecules such as VTN[4], highlighting the fundamental role of the adhesion machinery in promoting stemness. We recently described that hPSC colonies

**Fig. 4** Vinculin distribution and orientation within cornerstone adhesions. **a** Schematic representation of the different FA layers previously identified using iPALM in U2OS cells, colour bars highlight the $z$ range for each of the three layers and the force transduction layer containing vinculin and talin is emphasised over other layers. **b** iPALM analysis of the $Z_{centre}$ for N-terminally (Talin-1-N, $n = 54$ biologically independent ROI) and C-terminally (Talin-1-C, $n = 8$ biologically independent ROI) tagged (Eos) talin-1 in hPSC cornerstone adhesions. **c** iPALM analysis of the $Z_{centre}$ of N-terminally (Vinculin-N, $n = 55$ biologically independent ROI) and C-terminally (Vinculin-C, $n = 27$ biologically independent ROI) tagged (Eos) vinculin in hPSC cornerstone adhesions. **d** Two-colour iPALM images of Eos-tagged vinculin (Vinculin-C) and endogenous paxillin in a cornerstone FA. Individual cornerstone FA are displayed. Where localisation of vinculin and paxillin are displayed separately, top-view and side-view images are colour-coded as a function of the $z$-position of the indicated protein. Where fluorescence channels are merged (paxillin, red; vinculin, green), the $z$ position is only displayed in the side view and the colours represent the fluorescence signal for each protein. Scale bar 1 µm. **e** 3D scatter plots displaying the individual iPALM localisations (grey dots) of endogenous paxillin and Eos-tagged Vinculin-N and Vinculin-C within a single cornerstone adhesion. Surface plots present the fit of those localisations using a two-dimensional polynomial equation. Note that the paxillin localisations are homogeneously flat while localisations of both vinculin constructs form a solid paraboloid. **f** iPALM $xy$ images of Eos-vinculin-C at selected $z$-layers in an individual cornerstone FA. Scale bar 1 µm. Source data are provided as a Source Data file. Boxes display the median, plus the 1st and 3rd quartiles (IQR: 25th–75th percentile). Whiskers correspond to the median $\pm 1.5 \times$ IQR

---

are encircled by a strong contractile actin fence connected by exceptionally large cornerstone FA[19]. These adhesions are aligned to the colony edge and exert high traction forces upon the substrate[19]. Here, we investigated their 3D architecture using superresolution microscopy.

Our detailed analyses of cornerstone FA, using multiple microscopy approaches, revealed the unique properties of these adhesions. Using live-cell imaging of endogenously tagged paxillin we demonstrate remarkably slower dynamics, and thus higher stability, of cornerstone FA compared to FA formed at the centre of hPSC colonies. Superresolution iPALM imaging of core FA proteins revealed well-organised vertically demarcated, functional layers within hPSC cornerstone FA, congruent with classical FA[13]. However, while integrins β5 and αV, and paxillin demonstrated a $z$ position indicating a close relationship with the cell membrane and a vertical distribution very similar to those reported for other cell lines[13–15], the talin-1 and vinculin $z$ positions were found to be higher[13] (Fig. 7). Another feature that distinguishes cornerstone adhesions from classical FA is the unexpected upside down, head above tail, orientation of vinculin. Interestingly, a similar inverted vinculin orientation was also observed in mouse embryonic stem cells plated on laminin, but not when adhering to gelatin or fibronectin[27]. The functional significance of this inverted vinculin remains to be elucidated. In particular, it would be interesting to investigate whether this peculiar vinculin orientation is a feature of the pluripotent state of these cells and/or is associated with a specific integrin heterodimer/ECM combination. Moreover, unlike classical FA, cornerstone FA often featured a bi-phasic actin-regulatory layer (defined by actin and α-actinin-1), where two discrete vertical actin layers are separated by a less actin-dense 50 nm gap. This gap may constitute a functional layer in its own right and future work will aim at identifying which proteins localise here and how these two actin layers are regulated and structurally organised.

Here, we also determined the 3D organisation of kank1 and kank2 with respect to the core adhesion. As previously reported for classical FA[22,23], both kank1 and kank2 were laterally distributed around cornerstone FA. However, 3D localisation analyses of kank1 and kank2 revealed that they assemble strong FA-edge-defining vertical walls at the outer rim of paxillin-positive FA rather than a belt[22,23]. The vertical positioning of kanks around FA indicate that they are in close proximity to the force transduction layer, which is consistent with the ability of kanks to bind to talin[23]. However, further away from the adhesions both kank1 and kank2 localised vertically much lower, suggesting that recruitment of kanks to FA edges is distinct to their assembly in extended membrane proximal protein assemblies outside of FA. Functionally, kanks regulate the colony centre small FA distinctly from the colony edge cornerstone FA. Kank silencing induced

larger and more prominent FA in the colony centre, which is in line with previous work suggesting that in fibroblasts kank2 functions to limit integrin activity and facilitate adhesion sliding[22]. It is therefore extremely intriguing that the loss of kanks triggers the opposite outcome at the colony edge, reducing cornerstone FA size. Perhaps kanks represent a mechanism that allows hPSC colonies to distinguish between colony edge and centre cells. This concept and the regulatory pathways recruiting kanks to the different hPSC adhesions remain to be investigated.

Detailed imaging also revealed a lateral segregation of specific proteins within the cornerstone adhesions. In particular, β5 integrin was found to assemble high-density ring-like clusters at the edges of adhesions, while β1 and αV integrins were found to be more homogeneously distributed. Within classical FA, distinct integrin heterodimers display different dynamics[32] or cluster differently depending on their activation state[33]. However, the spatial segregation of specific integrin heterodimers appears to be a unique feature of cornerstone adhesions. This interesting centre-edge segregation within cornerstone FA was also observed with other FA components. Talin density was highest at the FA edges and the vertical distribution of vinculin was higher at FA edges compared to the centre. This apparent intra-FA organisation could be linked to the exceptionally large size of the cornerstone FA and specific interactions of FA components with proteins localising to the outer rim of the FA (FA belt) such as kanks[22,23]. Altogether these data indicate that the edges of cornerstone adhesions are structurally distinct from the centre and it is tempting to speculate that the edge and the centre may serve different biological functions, acting either as separate entities or working in synergy to maintain pluripotency of hPSC colonies.

## Methods

**Reagents antibodies and compounds.** Mouse primary antibodies used in this study were against paxillin (BD biosciences, 612405, 1:1000 for western blotting (WB), 1:100 for immunofluorescence (IF)), β-actin (Sigma Aldrich, Clone AC-15, Cat. No. A1978, 1:1000 for WB), GAPDH (Hytest, 5G4MaB6C5, 1:5000 for WB), Sox2 (R&D systems, MAB2018, 1:1000 for WB), vinculin (Sigma Aldrich, V9131, 1:1000 for WB, 1:100 for IF), talin (Sigma Aldrich, clone 8D4,T3287, 1:1000 for WB, 1:100 for IF), active β1 integrin (clone 12G10, in-house production), total β1 integrin (clone P5D2, in-house production), β3 integrin (AbD Serotec, MCA728, 1:100 for IF), αVβ5 integrin (Millipore, MAB2019Z, 1:100 for IF), αV integrin (clone L230, in-house production), paxillin (Santa Cruz, sc-365379, for iPALM samples 1:100), e-cadherin (Abcam, ab1416, 1:100 for IF), and α-actinin (Sigma Aldrich, A5044, 1:100 for IF). Rabbit primary antibodies used in this study were against Oct3/4 (Santa Cruz, sc-9081, 1:1000 for WB, 1:100 for IF), kank1 (Bethyl, A301-882A, 1:1000 for WB, 1:100 for IF), kank2 (Sigma aldrich, HPA015643, 1:1000 for WB), integrin β5 (Cell Signaling Technology, 3629, 1:100 for IF), e-cadherin (Cell Signalling Technology, 3195S, 1:1000 for WB) and paxillin (Santa Cruz, sc-5574, 1:100 for IF and for iPALM). The goat anti-Nanog antibody was from R&D systems (AF1997, 1:100 for IF). The secondary antibodies used in this study were all from Thermo Fisher Scientific and used at 1:300. These antibodies included Alexa Fluor 488 donkey anti-rabbit IgG, Alexa Fluor 488 donkey anti-mouse IgG, Alexa Fluor 568 donkey anti-rabbit IgG, Alexa Fluor 568 donkey anti-mouse IgG, Alexa Fluor 647 donkey anti-goat IgG, and Alexa

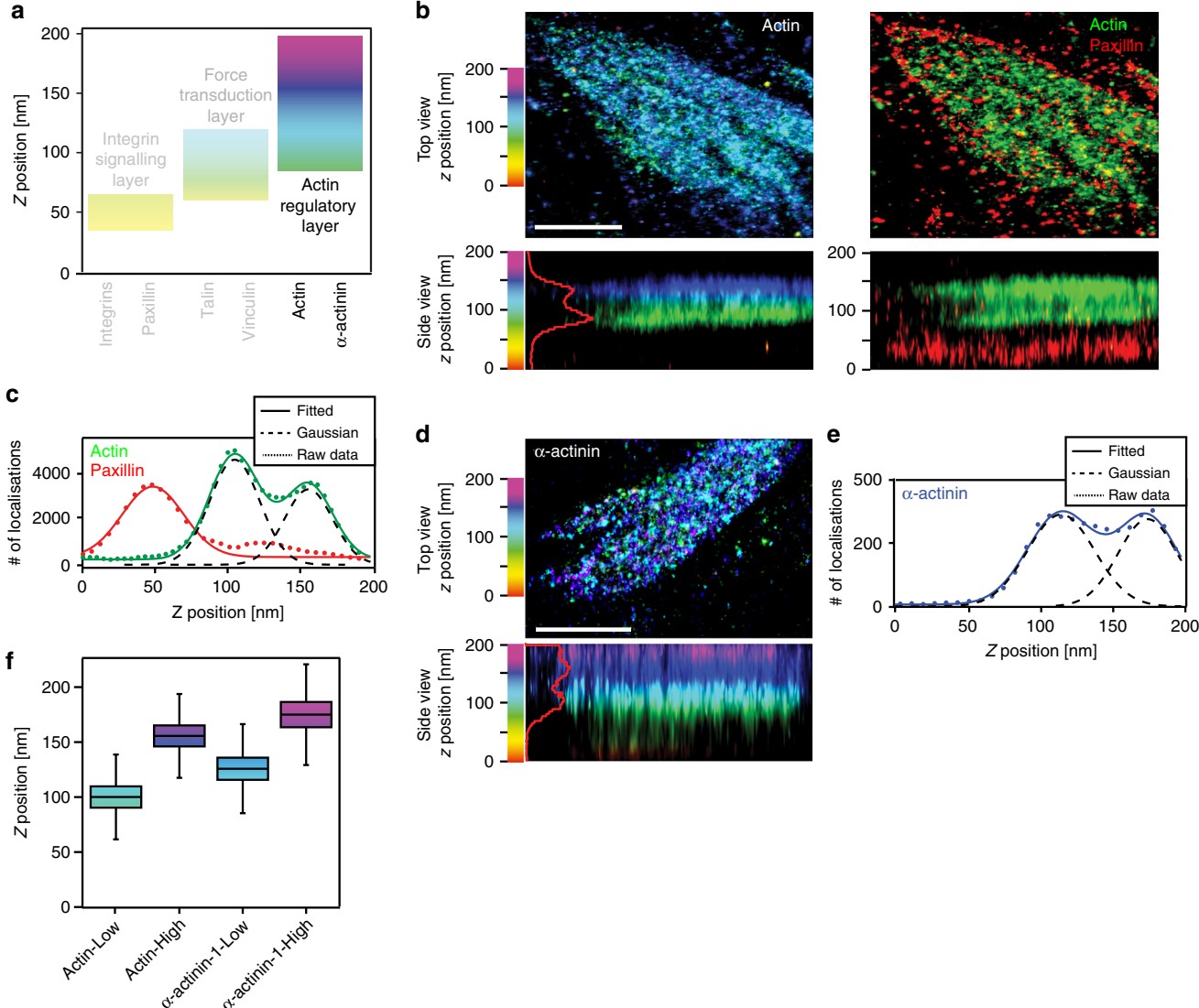

**Fig. 5** Actin and α-actinin-1 lateral distribution. **a** Schematic representation of the different FA layers previously identified using iPALM in U2OS cells, colour bars highlight the z range for each of the three layers and the actin-regulatory layer containing actin and α-actinin-1 is emphasised over other layers. **b** Two-colour iPALM images of Eos-tagged actin and endogenous paxillin in a cornerstone FA. One individual cornerstone FA is displayed. Where localisation of actin is displayed separately, top-view and side-view images are colour-coded as a function of the z-position of the actin molecules. Where fluorescence channels are merged (paxillin, red; actin, green), the z position is only displayed in the side view and the colours represent the fluorescence signal for each protein. Scale bar 1 μm. **c** Z density profile of paxillin (red) and actin (green) displaying the number of localisations as a function of the z position in an individual cornerstone adhesion. Dotted lines correspond to the experimental data, while solid lines correspond to the fitted data obtained using either a single Gaussian distribution (paxillin) or a sum of two Gaussian distributions (actin). Dashed black lines highlight these two Gaussian distributions. **d** iPALM image of Eos-tagged α-actinin-1 in an individual hPSC cornerstone FA. Top-view and side-view images are colour-coded as a function of the z-position of the α-actinin-1 molecules. Scale bar 1 μm. **e** Z-density profile of α-actinin-1 (purple) showing the number of localisations as a function of the z position. Dotted line corresponds to the experimental data while the solid line corresponds to the fitted data obtained using a sum of two Gaussian distributions (dashed black lines). **f** iPALM analysis of the $Z_{centre}$ of actin (actin low $n = 28$ biologically independent ROIs, actin high $n = 47$ biologically independent ROIs) and α-actinin-1 (α-actinin high $n = 31$ biologically independent ROIs, α-actinin low $n = 27$ biologically independent ROIs) in hPSC cornerstone adhesions. "Low" and "High" denote separate peaks in the distribution of the same protein. Source data are provided as a Source Data file. Boxes display the median, plus the 1st and 3rd quartiles (IQR: 25th–75th percentile). Whiskers correspond to the median ± 1.5 × IQR

Fluor 647 goat anti-mouse IgG. SiR-actin was provided by Cytoskeleton (Cat. No. CY-SC001). Atto 405 phalloidin and Alexa Fluor 488 phalloidin were provided by Thermo Fisher Scientific.

**Plasmids.** The following plasmids were provided by Addgene (gift from, Addgene plasmid number): tdEos-Talin-18 (Michael Davidson, 57672)[13], tdEos-Talin-N-22 (Michael Davidson, 57673)[13], tdEos-Vinculin-14 (Michael Davidson, 57691)[13], mEos2-Vinculin-N-21 (Michael Davidson, 57439), tdEos-Lifeact-7 (Michael Davidson, 54527)[13], mEos2-Actin-7 (Michael Davidson, 57339)[13], mEos2-Alpha-Actinin-19 (Michael Davidson, 57346)[13], tdEos-Paxillin-22 (Michael Davidson, 57653), pSpCas9(BB)-2A-GFP (Feng Zhang, 48138)[34], AICSDP-1:PXN-EGFP (The Allen Institute for Cell Science, 87420)[35].

The plasmid encoding β5-integrin-mEOS2 was a kind gift from Martin Humphries (University of Manchester, UK). GFP-kank1 and GFP-kank2 plasmids were gifts from Reinhard Fässler (Max Planck Institute of Biochemistry, Martinsried, DE)[22].

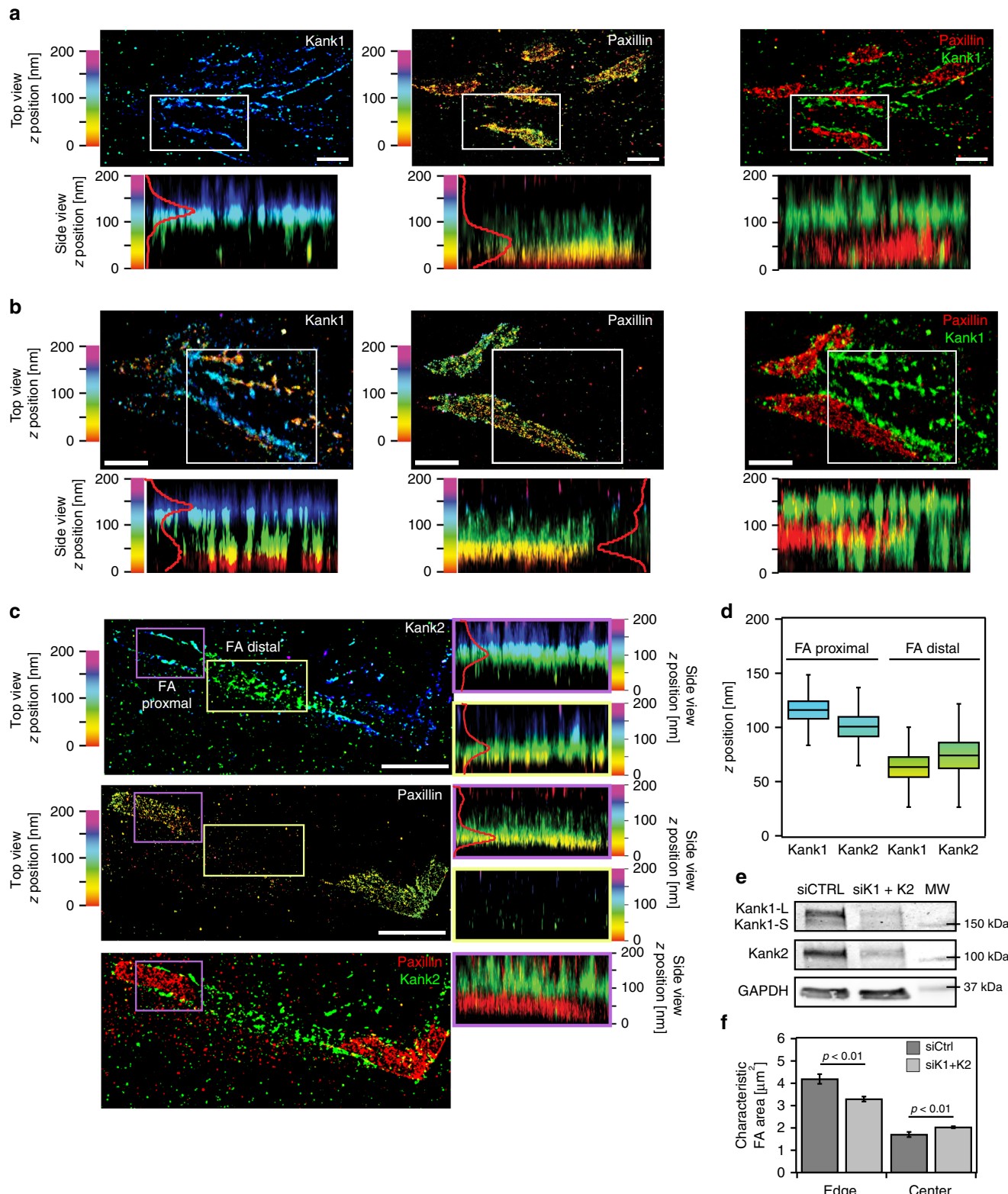

**Cell culture gene silencing and transient transfection**. The human-induced pluripotent stem cell (hPSC) lines HEL24.3 and HEL 11.4 were a kind gift from Timo Otonkoski (University of Helsinki, Finland). Cell lines were created using Sendai viruses[36,37]. Cells were grown in feeder-free conditions on Matrigel (MG) (Corning, 354277) or 5 µg/ml VTN (Life Technologies, A14700) in Essential 8 (E8) Basal medium (Life Technologies, A15169-01) supplemented with E8 supplement

(Life Technologies, A1517-01) at +37 °C, 5% $CO_2$ in a humidified incubator. Culture media was changed daily. For passaging, cells were detached using sterile-filtered 0.5 mM EDTA (Life Technologies, 15575-038) in PBS for 3 min at room temperature (RT)[19]. U2OS osteosarcoma cells here were grown on VTN-coated plates in E8 media. U2OS cells were purchased from DSMZ (Leibniz Institute DSMZ-German Collection of Microorganisms and Cell Cultures, Braunschweig DE,

**Fig. 6** Nanoscale 3D localisation of kank1 and kank2 in hPSC. **a–c** Two-colour iPALM images of endogenous paxillin together with Eos-tagged kank1 (**a**, **b**) or kank2 (**c**) in the vicinity of hPSC cornerstone adhesions. Multiple cornerstone FA are displayed. Kank1 forms a wall surrounding cornerstone adhesions (**a**) and is also found outside of adhesions (**b**). Kank2 also localises both in proximity (**c**, purple box), and distal (**c**, yellow box), to cornerstone adhesions. Where localisation of kank1 and kank2 are displayed separately, top-view and side-view images are colour-coded as a function of the z-position of the indicated molecules. Where fluorescence channels are merged (paxillin, red; kank1/kank2, green), the z position is only displayed in the side view and the colours represent the fluorescence signal for each protein. Side views are for selected regions (white, yellow, and purple insets). Scale bars 1 μm. **d** iPALM analysis of the $Z_{centre}$ of kank1 (Kank1 adjacent, $n = 29$ biologically independent ROIs; Kank1 distal, $n = 17$ biologically independent ROIs) and kank2 (Kank2 adjacent, $n = 17$ biologically independent ROIs; Kank2 distal, $n = 15$ biologically independent ROIs) when in close proximity (adjacent) to cornerstone adhesions or when distant to cornerstone adhesions. **e** Western blot analysis of kank1 and kank2 proteins levels in hPSC pretreated with either control siRNA (siCTRL) or a combination of siRNA targeting kank1 and kank2 (siK1 + siK2) ($n = 3$). **f** Quantification of the characteristic FA area at the edges or in the middle (centre) of hPSC colonies pretreated with siCTRL or siK1 + siK2 ($n = 3$ biologically independent experiments). Bars represent the characteristic FA size obtained by fitting a weighted sum of two exponential densities to a histogram of FA area distribution. Statistics: Student's t-test. Error bars depict one standard deviation error in the fit. Source data are provided as a Source Data file. Boxes display the median, plus the 1st and 3rd quartiles (IQR: 25th–75th percentile). Whiskers correspond to the median ± 1.5 × IQR

ACC 785). ARPE-19 normal retinal epithelial cells were a kind gift from Olav M. Andersen (Aarhus University, Denmark) and were cultured in Dulbecco's modified Eagle's medium: nutrient mixture F12 (DMEM/F12) (Life Technologies, 10565-018) supplemented with 10% foetal bovine serum (Biowest, S1860) at +37 °C, 5% $CO_2$ in a humidified incubator.

To induce stem cell differentiation, hPSC were cultured on VTN-coated plates and subsequently differentiated using a mixture of Essential 6 (E6) (Gibco, A1516401) media and 100 ng/ml BMP-4 (R&D systems, 314-BP). The differentiation media was changed daily and the samples were collected after 1, 3 and 6 days.

Plasmids of interest were transfected using DNA-in reagent (MTI-Global Stem) according to the manufacturer's instructions. 24 h before transfection, the cells were seeded on MG to reach appropriate cell confluence of 80%. DNA-in reagent and plasmid DNA were mixed in a ratio of 1:3 (μg of DNA:μl of regent), in Opti-MEM-reduced serum medium (Gibco, 31985070), 15 min before the transfection. Cells were gently detached using 0.5 mM EDTA in PBS, diluted in E8 and mixed with transfection mixture with gentle pipetting. The cell-transfection mixture was then added to VTN-coated glass and incubated for 20–24 h.

To transiently suppress kank1 and kank2 expressions in hPSC, Accell siRNA pools (Dharmacon, kank1 E-012879-00-0005; kank2 E-027345-00-0005) were used according to the manufacturer's protocol. siRNA were diluted to a final concentration of 1 μM. siRNA transfection was done first in suspension during the cell passage and repeated after 24 h on adherent cells. 48 h after the initial transfection, cells were plated on VTN-coated plates and grown for 24 h before fixation or lysis to collect protein samples. siRNA used as a control was Accell Non-targeting Pool (Dharmacon, D-001910-10-20).

To stably suppress vinculin expression, hPSC were transduced with shRNA containing viral particles in suspension during the cell passage. Puromycin selection (2 μg/ml) was started 48 h after the transduction. The shRNA used as control (shCTRL) was shScramble provided by Sigma Aldrich (SHC002, sequence: CCG GCA ACA AGA TGA AGA GCA CCA ACT CGA GTT GGT GCT CTT CAT CTT GTT GTT TTT). The shRNAs targeting vinculin shVin#1 (TRCN0000116755, sequence: CCG GCG GTT GGT ACT GCT AAT AAA TCT CGA GAT TTA TTA GCA GTA CCA ACC GTT TTT G) and shVin#2 (TRCN0000116756, sequence: CCG GGC TCG AGA TTA TCT AAT TGA TCT CGA GAT CAA TTA GAT AAT CTC GAG CTT TTT G) were provided by the functional genomics unit of the University of Helsinki.

**Generation of tagged kank1 and kank2 tdEos constructs**. XhoI/EcoRI restriction sites were introduced to flank both mouse kank1 and kank2 genes using polymerase chain reaction (PCR) (Phusion hot start II polymerase, Thermo Fisher Scientific). Primers used were kank1_For (5′-AAT ACT CGA GAT GGC TTA TAC CAC AAA AGT TAA TG-3′) kank1_Rev (5′-AAT AGA ATT CCG TCA AAA GAA CCT CGG TGA G-3′), kank2_For (5′-AAT ACT CGA GAT GGC CCA GGT CCT GC-3′) and kank2_Rev (5′-AAT AGA ATT CCC TCC TCG GCT GAA GAC GA-3′). Following PCR amplification, kank1 and kank2 sequences were inserted into tdEos backbone (gift from Michael Davidson, tdEos-N1, Addgene plasmid no. 54634) using XhoI/EcoRI restriction sites. The constructs were sequenced to validate integrity.

**Generation of a paxillin-GFP cell line**. To generate the endogenously tagged paxillin HEL24.3 hPSC line [3535] the gRNA sequence targeting paxillin (5′-GCACCTAGCAGAAGAGCTTG-3′) was introduced into pSpCas9(BB)-2A-GFP backbone using the BbsI-restriction site. The template plasmid (AICSDP-1:PXN-EGFP) was provided by Addgene[35]. HEL 24.3 hPSC were then transfected with the GFP-Cas9-paxillin_gRNA construct and template plasmid (AICSDP-1:PXN-EGFP) in equimolar ratio (1:1). After transfection, the cell line was cultured for 5 days before sorting the cells based on green fluorescence with FACS (FACSAria IIu, BD).

**Nano-grid experiments**. hPSC were seeded on 35 mm nano-grid surface topography glass dishes (Nanosurface Biomedical) coated with 5 μg/ml VTN. The nanopattern topography consisted of several parallel 800 nm wide ridges interspersed with 600 nm deep grooves (termed nano-grids). Cells were cultured on nano-grids for 24 or 72 h for microscopy in either in E8 or E6 basal medium. For protein analysis samples were collected on day 1, day 3, and day 6 E6 medium. Samples were then acquired either fixing the cells with 4% PFA in PBS for microscopy or lysing the cells for protein analysis.

To investigate the recovery of the pluripotency factors, the hPSC colonies were seeded either on VTN-coated uniform surfaces and grown in E8 medium for 3 days or on VTN-coated nano-grids and grown in E6 medium for 3 days to induce rapid differentiation. Subsequently the cells from both conditions were passaged to uniform VTN-coated surface in the presence of E8 medium for additional 3 days prior to protein sample collection.

**SDS–PAGE and quantitative western blotting**. Protein extracts were separated under denaturing conditions by SDS–PAGE and transferred to nitrocellulose membranes. Membranes were blocked for 1 h at RT with blocking buffer (LI-COR Biosciences) and then incubated overnight at 4 °C with the appropriate primary antibody diluted in blocking buffer. Membranes were washed with PBS and then incubated with the appropriate conjugated secondary antibody diluted 1:5000 in blocking buffer for 2 h. Membranes were washed in the dark and then scanned using an Odyssey infrared imaging system (LI-COR Biosciences) or incubated with ECL Plus Western blotting reagent (GE Healthcare) and the photographic film developed. Band intensity was determined by digital densitometric analysis using Fiji software[38]. The most important blots are provided in the Source Data file.

**Immunofluorescence**. Cells were seeded on VTN-coated glass-bottom microscopy dishes for 20–24 h, washed twice with PBS and simultaneously fixed and permeabilized using 4% PFA and 0.3% Triton-X-100 in PBS for 15 min. Cells were then washed with PBS and the fixative quenched with 0.1 M glycine for 10 min. following quenching, cells were washed and incubated with primary antibodies diluted in 1% BSA in PBS for either 2 h at RT or at +4 °C overnight. The samples were then washed twice with PBS, incubated with secondary antibodies diluted in 1% BSA in PBS for 2 h at RT, washed with 0.2% Tween 20 for 10 min at RT followed by a final PBS wash.

**Microscopy**. The confocal microscope used was a laser scanning confocal microscope LSM880 (Zeiss) equipped with an Airyscan detector (Carl Zeiss). Objectives used were ×40 water (NA 1.2) and ×63 oil (NA 1.4). The microscope was controlled using Zen Black (2.3) and the Airyscan was used in standard super-resolution mode.

The spinning disk microscope used was a Marianas spinning disk imaging system with a Yokogawa CSU-W1 scanning unit on an inverted Zeiss Axio Observer Z1 microscope controlled by SlideBook 6 (Intelligent Imaging Innovations, Inc.). Objectives used were a ×20 (NA 0.8 air, Plan Apochromat, DIC) objective (Zeiss), a ×63 oil (NA 1.4 oil, Plan-Apochromat, M27 with DIC III Prism) objective (Zeiss), or a ×100 (NA 1.4 oil, Plan-Apochromat, M27) objective. Images were acquired using an Orca Flash 4 sCMOS camera (chip size 2048 × 2048; Hamamatsu Photonics).

The structured illumination microscope (SIM) used was DeltaVision OMX v4 (GE Healthcare Life Sciences) fitted with a ×60 Plan-Apochromat objective lens, 1.42 NA (immersion oil RI of 1.514), used in SIM illumination mode (five phases × three rotations). Emitted light was collected on a front illuminated pco.edge sCMOS (pixel size 6.5 μm, readout speed 95 MHz; PCO AG) controlled by SoftWorx.

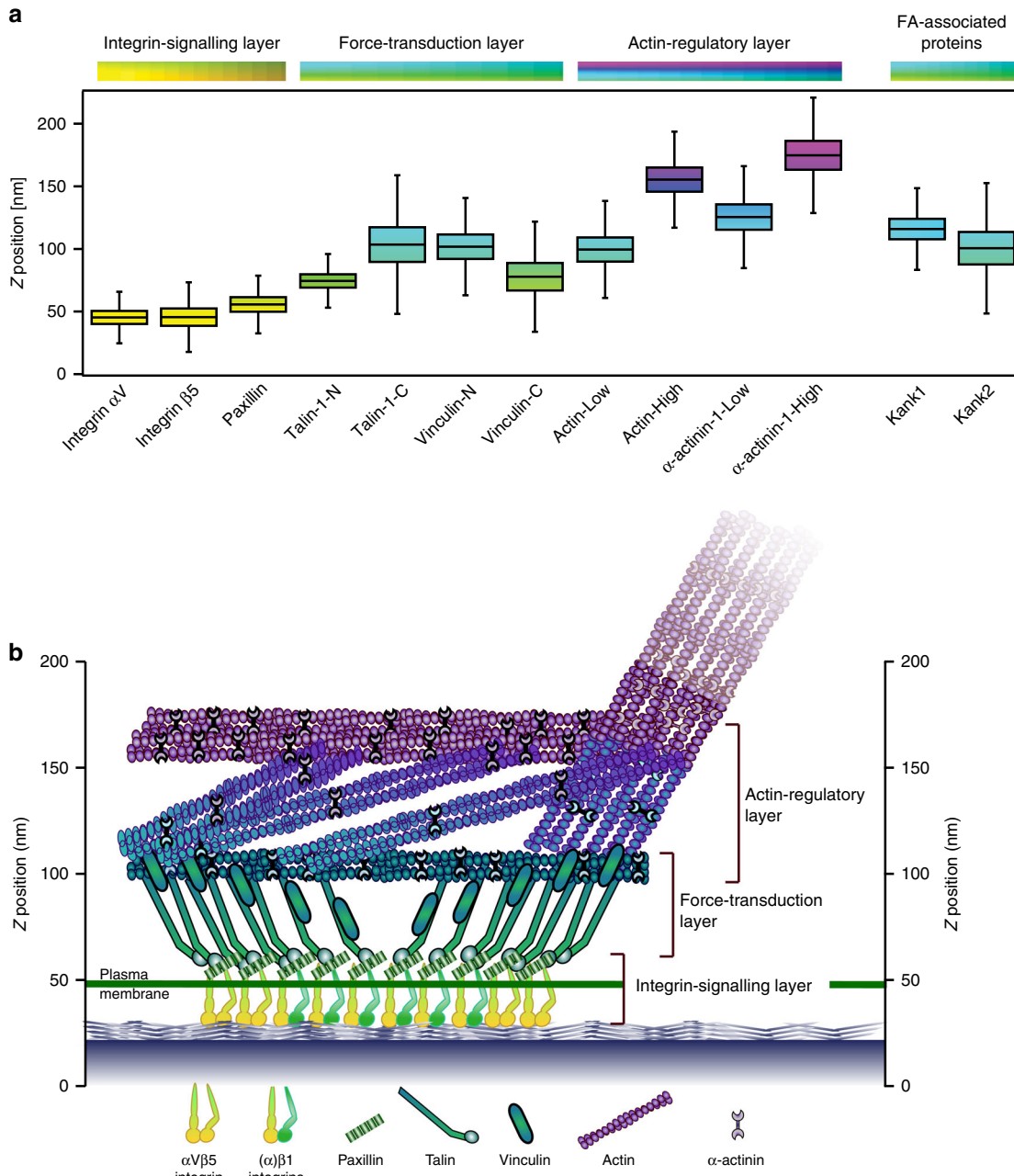

**Fig. 7** Overall vertical stratification of cornerstone FA. **a** Z positioning ($Z_{centre}$) of αV and β5 integrins, paxillin, vinculin, talin-1, actin, α-actinin-1, kank1 and kank2 in cornerstone adhesions. Letters –N and –C denote the location of the tag while "low" and "high" denote separate peaks in the distribution of the same protein. Boxes display the median, plus the 1st and 3rd quartiles (IQR: 25th– 75th percentile). Whiskers correspond to the median ± 1.5 × IQR. **b** Schematic model of the 3D architecture of hPSC cornerstone adhesions. The lateral and vertical positioning of each protein are based on the data presented here. This model does not depict protein stoichiometry. Source data are provided as a Source Data file

**Paxillin dynamics experiments**. hPSC paxillin-EGFP cells were seeded on VTN-coated glass-bottom dishes (Cellvis, D35-10-1.5-N) for 15–18 h. Cells were then imaged live using a Marianas spinning disk imaging system with a Yokogawa CSU-W1 scanning unit on an inverted Zeiss Axio Observer Z1 microscope controlled by SlideBook 6 (Intelligent Imaging Innovations, Inc.). Images were acquired every minute using a Photometrics Evolve, 10 MHz Back Illuminated EMCCD (512 × 512 pixels) camera and ×40 Zeiss LD C-Apochromat water objective (NA1.1).

To quantify the difference in behaviour between the cornerstone adhesions (found on the colony edge) and the FA found in the centre of the hPSC colonies, the edge and the centre of the colonies were separated. The edge was defined as a 5 μm-thick strip starting from the line delimiting the colony boundary and extending towards the colony centre. The centre was defined as the remaining region of the colony. The FA present at the colony edge or centre were first segmented using the

Trainable WEKA segmentation plugin[39] implemented within Fiji[38]. Movies were then further processed using the Focal adhesion analysis server (http://faas.bme.unc.edu/)[40] to compute the lifetime and maximal size of all paxillin-positive adhesions. For analysis purposes, the maximum adhesion lifetime was set at 105 min to match the length of the shortest live-imaging replicate experiment. Data were binned as a function of maximal size or lifetime using GraphPad Prism 7.01 software.

**iPALM**. iPALM imaging was performed on[13,16] cells plated on 25 mm diameter round coverslips containing gold nanorod fiducial markers[13,16] (Nanopartz, Inc.), passivated with a ca. 50 nm layer of $SiO_2$, deposited using a Denton Explorer vacuum evaporator. After fixation, an 18 mm coverslip was adhered to the top of

the sample and placed in the iPALM. Eos-tagged samples were excited using 561 nm laser excitation at ca. 1–2 kW/cm$^2$ intensity in TIRF conditions. Photoconversion of Eos was performed using 405 nm laser illumination at 2–10 W/cm$^2$ intensity. 40,000–80,000 images were acquired at 50 ms exposure, and processed/localised using the PeakSelector software (Janelia Research Campus). Alexa Fluor 647-labelled samples were imaged similarly, but with 2–3 kW/cm$^2$ intensity 647 nm laser excitation, and 30–40 ms exposure time in STORM-buffer containing TRIS-buffered glucose, glucose oxidase, catalase, and mercaptoethanol amine[41].

iPALM data were analysed and images rendered using the PeakSelector software (Janelia Research Campus)[13,16]. iPALM localisation data records both the fluorescent molecules localised within the FA, as well as molecules in the cytoplasmic fraction, to quantify the spatial distribution of the proteins within individual FA, we zoomed into areas covered only by the FA and the immediate surrounding space. To render iPALM images, a single colour scheme was used from red to purple, covering the $z$ range 0–200 nm, where features within FA are seen. The same colour scheme was also used for side-view ($xz$) images. For analysis of protein distributions in FA: $Z_{centre}$ and svert calculation, the three-dimensional molecular coordinates for each region (individual FA) were analysed to obtain histograms of vertical positions with 1-nm bins. The centre vertical positions ($Z_{centre}$) and width (svert) were determined from a Gaussian fit to the FA molecule peak. For proteins like actin and α-actinin, where dual peaks were observed, the fitting was done using the sum of two-Gaussian distributions with independent centre vertical position and width. After the histograms for all images and individual FA were obtained, they were combined into a single average $Z_{centre}$ and svert.

**Quantification of focal adhesion properties**. For the quantification of FA area, FA number, density, and characteristic area were analysed using the ImageJ software (NIH)[19]. Individual FA were first isolated and measured using the 'analyze particles' built-in function of ImageJ. Second, as we observed that the distribution of FA area was not Gaussian but rather a heavy-tailed distribution, this was approximated, and fitted in Igor Pro (WaveMetrics) to a hyper-exponential distribution or, in other words, a weighted sum of two exponential densities. Finally, the fit provided two characteristic areas of FA for each curve, A1 describing the smaller FA and A2 describing the larger FA. As A1 was not significantly different between samples, we focused on the differences found on A2 that describes the larger FA rarely present in fibroblasts or at the centre of hPSC colonies, but regularly present at the edge of hPSC colonies. For simplicity, throughout the paper the characteristic area A2 is referred to as 'characteristic FA area'. In the case of hPSC colonies, as there were clear visible differences in the FA size at the edge of the colony, we separated the edge from the centre for quantification. The edge was defined as a 40-pixel-thick strip starting from the line delimiting the colony boundary and extending towards the inside of the colony. The centre was defined as the remaining region of the colony.

**Statistical analysis**. If not indicated otherwise, statistical analyses were performed using either a Student's $t$ test (when comparing pairs of data) or an analysis of variance (ANOVA, when comparing more than two groups) complemented by Tukey's honest significant difference test (Tukey's HSD). The software R version 3.3.3 (R Development Core Team, Vienna, Austria) was used to perform these analyses. Statistical significance levels are annotated as ns = non-significant ($p > 0.05$) or by providing the range of the $p$-value.

**Ethical considerations**. The hPSC lines have been derived under the approval of the Coordinating Ethics Committee of Helsinki and Uusimaa Hospital District (statement no. 423/13/03/00/08).

**Reporting summary**. Further information on research design is available in the Nature Research Reporting Summary linked to this article.

## Data availability

All data supporting the findings of this study are available within the paper and its supplementary information files, or from the corresponding author on reasonable request. The source data underlying Figs. 1d–f, 2e, 3f, 4b, c, 5f, 6d–f, and Supplementary Figs. 1c and g, 2, 9b and d and 10b are provided as a Source Data file.

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

## Acknowledgements

We thank J. Siivonen and P. Laasola for technical assistance. H. Hamidi is acknowledged for editing the manuscript and the figures. The Ivaska lab is acknowledged for lively discussions and critical feedback on the manuscript. We thank Timo Otonkoski, Olav Andersen, Michael Davidson, Feng Zhang, Reinhard Fässler, Janet Askari, Martin Humphries and The Allen Institute for Cell Science for providing reagents. The Cell Imaging and Cytometry Core (Turku Bioscience Centre, University of Turku, Åbo Akademi University and Biocenter Finland) and the functional genomics unit of the University of Helsinki (Research Programs Unit, HiLIFE Helsinki Institute of Life Science, Faculty of Medicine, University of Helsinki, Biocenter Finland) are acknowledged for services, instrumentation, and expertise. This study has been supported by the Academy of Finland (G.J., E.N. 297079 and J.I.) and an ERC CoG grant 615258 (J.I.). A.S. has been supported by the University of Turku Doctoral programme for Molecular Medicine (TuDMM). iPALM microscopy was performed at the Howard Hughes Medical Institute (HHMI) Janelia Research Campus Advanced Imaging Center (AIC). The AIC is jointly sponsored by the HHMI and the Gordon and Betty Moore Foundation.

## Author contributions

Conceptualisation, J.I., C.G. and A.S.; Methodology, A.S., C.G., E.N., J.A., G.J.; Formal analysis, C.G., A.S., E.N., J.A.; Investigation, A.S., C.G., E.N., J.A., M.S., J.I.; Resources, M. M.; Writing—original draft, C.G., J.I., G.J.; Writing—review and editing, C.G., A.S., E.N., J.I., G.J.; Visualisation, C.G., A.S., G.J.; Supervision, J.I. and T-L.C.; Funding acquisition, J.I.

## Competing interests

The authors declare no competing interests.
