## [Peer Review File · Nature Communications]

Reviewers' comments:

Reviewer #1 (Remarks to the Author):

This study extends earlier work in the authors' laboratory on focal adhesions in human pluripotent stem cell (hPSC) colonies grown in vitro. The authors previously compared focal adhesions in hPSC to those in fibroblasts and found that hPSC colonies unlike fibroblasts were encircled by a large fence of focal adhesions reinforced by a belt of actin. In this study, the authors investigate the role of these structures in maintenance of pluripotency by disrupting them. They then apply super-resolution microscopy to describe these structures and their components in greater detail in three dimensions. Integrins, talin, vinculin, actin and the focal adhesion associated kank proteins were labelled and visualized at high resolution using the IPALM technique. These studies show that the structures around the edge of the colony are quite stable and reveal the stacked organization of the structures in the z-dimension. Distribution of the components of the focal adhesions showed some unique features relative to previous work with other cell types. Kank knockdown altered the pattern of focal adhesion size with colonies.

The application of state of the art imaging techniques to hPSC is certainly of real interest to the stem cell field and the study provides some unique structural perspectives on hPSC. However, there are some concerns that limit the significance of the observations reported in this study. These concerns relate to three main areas.

1. The experiment in Figure 1 is the only part of the study that addresses the role of these unusual focal adhesions in maintenance of pluripotency (the rest of the study is a high quality description of the focal adhesions at the structural level). However, there are aspects of the design of this experiment that raise questions. What exactly does culture on these nanogrids alter? For example it is well established through experiments with micropatterning that colony size per se on patterned substrates exercises constraints on self-renewal and differentiation. Do the colonies grow up the edges of the troughs, or are they confined within them, and if so, how does this work relate to previous studies with micropatterned hPSC culture? Moreover, it is not clear to me that other aspects of subcellular organization might not be disrupted by culture of hPSC on the grid structure. Mechanical forces that impact signal transduction in epithelia cells are mediated not only by interactions with the extracellular matrix through focal adhesions, but also through cell-cell junctions. It is well known that cell-cell contacts are essential for maintenance of pluripotency in hPSC. Importantly it is not clear whether cell-cell junctions are properly maintained on the grid under these conditions, or for that matter, if other parameters apart from focal adhesion size are not altered. Finally the endpoint is difficult to interpret. It seems likely that culture on the grid (whatever mechanism is invoked to account for its effects) is accelerating a differentiation process that will eventually occur in control cells as well in the absence of appropriate growth factors. This complex endpoint of accelerated differentiation (the authors studied only one time point which oversimplifies matters) could be modified by many other factors apart from focal adhesion size.
2. In supporting their claims to identify features of the focal adhesions in hPSC that appear to be unique, it is not clear to me whether the authors have carried out appropriate comparisons. hPSC display properties typical of epithelia, but the authors appear to compare them with fibroblasts and with osteosarcoma cells. It would be more informative to carry out these comparisons with a typical diploid cultured epithelial cell such as a keratinocyte. Then it would be more clear if the structure of focal adhesions reported here is really unique to pluripotent stem cells.
3. The experiments on kank knockdown are certainly interesting, in that they show a change in the size distribution of focal adhesions within a colony. However, it is curious that given the experiments in Figure 1, that the authors did not further investigate the effects on kank knockdown on cell differentiation. (Such experiments would also have to take into account possible concomitant effects on cell cell adhesion). It would seem that this would be a more direct approach to investigating the role of the large focal adhesions in maintaining pluripotency than the experiments with the nanogrids.

Reviewer #2 (Remarks to the Author):

The paper reports 3 main findings: 1) using nanopatterns to demonstrate that the physical restriction of adhesion size in hPSC colonies is sufficient to trigger differentiation; 2) dissect the nano-structure of these cornerstone adhesions that are gatekeepers for pluripotency; 3) depletion of kinks diminishes the patterning of the cornerstone adhesions. While defining and characterizing cornerstone adhesions and their relationship to pluripotency is novel and significant, there are numerous issues that need to be addressed before this paper could be recommended for publication. Some of the discussion points are not supported by data included in the paper. Either data needs to be added or the discussion needs to be modified. Similarly, throughout the text information is presented that does not seem to connect to the story being presented, but raise questions. Again, these should be addressed or eliminated (eg., location of activated beta1 integrin – why is this the only integrin whose activity was tested)? Finally, as documented below, the colors in the figures, and the discrepancies between Figs 6a and 6b need to be addressed.

Specific points:

It seems that the cornerstone adhesions are structurally and functionally distinct from focal adhesions. Therefore, referring to them as unusually large focal adhesions is confusing, and flipping back and forth between terms makes reading difficult and reduces the impact of the finding. Would recommend using cornerstone adhesion and clearly establishing the definition of this type of adhesion with the data presented in the paper. However, this raises the question of why the particular molecules were investigated with iPALM.

A previous paper from this group (Närvä et al, 2017) states that cell-cell junctions (E-Cadherin) contributed to actin fence integrity. Why wasn't E-Cadherin investigated with iPALM? Or, one of the other modes of microscopy used in the paper?

The explanation of the vinculin results is highly speculative. The Case and Xia references use mutants to support their claims. Moreover, the possible interaction of vinculin with alpha-actinin was given as an explanation for the "inverted orientation" of vinculin in the Xia paper, but it is not cited as such. Referencing should be more accurate and without mutant support, claims should be dialed back to match the level of included data.

If the speculation is that vinculin is interacting with alpha-actinin instead of talin, why are data (numbers) not included in the text? This was done with alpha-actinin and actin.

In the vinculin knockdown does the second layer of actin change?

If kinks regulate microtubule targeting, what happens to microtubules? Is there interplay between microtubules and the actin cables? If it is outside the scope, then clarify.

Cornerstone adhesion dynamics are not quantified. They are presented in a single movie, where the adhesions on the right seem much slower to turnover than the adhesions on the left. Quantification is needed for multiple experiments.

The discussion claims that the size and orientation of cornerstone adhesions are essential for pluripotency. This is not directly shown. While supplemental figures demonstrate that different perturbations that affect pluripotency decrease the size of cornerstone adhesions, or the density around the periphery, a direct causal relationship between them is not established. Therefore, a more careful wording is required.

No details are given about the reconstruction of single molecule images. What were the lateral and axial localization precisions? The very large clusters in Fig 2a suggest different localization precision than Fig 2b, and Fig 2c seems (by eye) different yet again.

Some of the data, vinculin and kinks, are claimed to have axial distributions that vary laterally. This is nicely demonstrated for vinculin in Fig 3e, one wonders why something similar was not done the kinks?

Is all of beta1 active? Why does it not localize with talin, one of its activators? Is the activity of

other integrins constrained?

Why wasn't the other VN receptor alpha v beta3 investigated? The paper mentions using beta 5 and alpha v beta 5 antibodies, why not alpha v beta 3 and alpha v?

In 5e is it focal adhesion or cornerstone adhesion area?

The term horizontal segregation seems inaccurate since there is no fixed coordinate system—spatial segregation to the periphery of adhesion would perhaps be clearer.

N analyzed and statistics used seem to be missing for the most part.

There are numerous issues with the figures that need to be addressed.

- o Why was Fig 2e repeated in next 3 figures? It took multiple tries to understand that the opacity change was to indicate which layer was being addressed in the rest of the figure.
- o The left-to-right shading of ALL of the box and whiskers does not seem to have any relevance to axial data. Why is it there? There is no explanation for it. Similarly, why are the bars in the repeating Fig 2e shaded left-to-right?
- o Why do the axial height color bars in ALL of the figures have left-to-right variations? It is especially obvious in the yellow region around 50 nm.
- o What are the boxes under the side views of adhesions in Figs 2a-c? Normalized intensity?
- o The right-hand panel of Fig 3d would be more informative as a Vinculin-N. It would match the panel below it, and avoid the confusing changes in color for paxillin and vinculin in Fig 3d.
- o Again, why does the color change between Fig 4b left and 4b right?
- o Why don't the layers in Fig 6b match the data in Fig 6a? For example, the first layer of actin is below 100 nm in Fig 6a and above 100 nm in Fig 6b.
- o Why weren't the elevations of proteins in the periphery shown in the cartoon?
- o Finally, the colors in Fig 6 do not readily distinguish the layers.

Reviewer #3 (Remarks to the Author):

. The authors have used super-resolution microscopy to examine the edges of hESC colonies that have unusual actin 'fences'. In the abstract and introduction, they indicate that the hypothesis is that the atypical Focal adhesions seen at the borders of hESC colonies are gatekeepers to pluripotency. If this is the big question that they sort to address in the paper, then the data do not answer it. They nicely characterise the 'cornerstone' adhesions by super resolution microscopy techniques and show some similarities/differences to some previous reports but it's not clear what that means for these cornerstone adhesions in terms of function – here in relation to pluripotency / fate determination signals. They do add in a new part in that no-one has done this sort of super-resolution microscopy for Kank before.

Overall, the study seems to have been done competently although I am not a super-resolution expert, but the major question is unclear and the hypothesis that these cornerstone adhesions are responsible for maintaining pluripotency is not really addressed. The paper seems to be reporting a number of interesting observations but neither linking them all or focussing on one question. How relevant their findings are since they use only 1 hESC line, perform limited nano-surface experiments and then do not link the 2 parts of the paper together is a question.

1. They are a reputable group with experienced imaging specialists and seem to be doing things the right way. I do worry, however, that they may be pushing the limits of the data interpretation. For example in figure 3 panel e they describe concave ('cup-like') adhesion structures. How can they be sure this is not an imaging artefact?

2. Figure 1 describes data to suggest that nano-grated surfaces alter the organisation of cornerstone adhesions resulting in altered cell fate responses. They don't have any evidence for this. How can they be sure that it is the altered cornerstone adhesions that result in the effects observed in panels d-f on Sox2 and SSEA-1 expression? Many alternative explanations for these effects are surely possible. For example, the pluripotent colonies comprise many cells with

numerous cell-cell interactions. Are there changes in cell-cell adhesions that cause these effects? Cell-cell adhesion changes are well known to drastically affect hESCs,

3. Furthermore, loss of two markers when cells were transferred into E6 (differentiation base medium without stem cell maintaining activin and FGF) cannot be constituted as loss of pluripotency and it is notable that Oct 4 and Nanog are not examined. Do these cells differentiate to progenitors quicker than with BMP? Importantly the authors did not look at transferring the colonies back into stem cell maintenance medium from E6...do they recover these 2 markers just as quickly. In mESC it has previously been shown by Chambers and colleagues that even Nanog can be lost for days without changing pluripotent status on recovery.

4. Figure 1 only describes one type of nano-grated surface. Why were these dimensions chosen? What happens when they are altered? A more detailed analysis of these parameters together with the resultant quantification of the cornerstone adhesions would benefit the study. Also how is the nanoscale architecture of the cornerstone adhesions altered by the nano-grated surface? Why isn't this linked and compared to the pluripotency marker results?

5. The whole paper seems to be performed with only one hESC stem cell line. This is not the standard for work with human pluripotent stem cells (with very different genetic background). How representative are these findings to other stem cell lines? Key experiments at minimum should be validated in a second line.

6. IF panels in figure 2 and onwards appear to be of individual cornerstone adhesions but this is not made clear in the manuscript. It should be clarified for the reader so that they can easily relate these images to the part of hESC colonies that is being shown.

7. Nick Brown in Cambridge has shown different orientations of talin by super resolution microscopy in flies. This paper could also be referenced and the findings discussed. Klapholz et al Current Biology Volume 25,

<https://www.sciencedirect.com/science/article/pii/S0960982215000755#sec2>

8. Vitillo and colleagues Stem Cell Reports 2016 also published data implicating FAK in pluripotent stem cell maintenance using a number of lines. This work should also be referenced

[https://www.cell.com/stem-cell-reports/fulltext/S2213-6711\(16\)30130-8](https://www.cell.com/stem-cell-reports/fulltext/S2213-6711(16)30130-8)

Reviewer #1 (Remarks to the Author):

This study extends earlier work in the authors' laboratory on focal adhesions in human pluripotent stem cell (hPSC) colonies grown in vitro. The authors previously compared focal adhesions in hPSC to those in fibroblasts and found that hPSC colonies unlike fibroblasts were encircled by a large fence of focal adhesions reinforced by a belt of actin. In this study, the authors investigate the role of these structures in maintenance of pluripotency by disrupting them. They then apply super-resolution microscopy to describe these structures and their components in greater detail in three dimensions. Integrins, talin, vinculin, actin and the focal adhesion associated kank proteins were labelled and visualized at high resolution using the IPALM technique. These studies show that the structures around the edge of the colony are quite stable and reveal the stacked organization of the structures in the z-dimension. Distribution of the components of the focal adhesions showed some unique features relative to previous work with other cell types. Kank knockdown altered the pattern of focal adhesion size with colonies.

The application of state of the art imaging techniques to hPSC is certainly of real interest to the stem cell field and the study provides some unique structural perspectives on hPSC. However, there are some concerns that limit the significance of the observations reported in this study. These concerns relate to three main areas.

We would like to thank the reviewer for these positive and encouraging comments and for the insightful suggestions.

1. The experiment in Figure 1 is the only part of the study that addresses the role of these unusual focal adhesions in maintenance of pluripotency (the rest of the study is a high quality description of the focal adhesions at the structural level). However, there are aspects of the design of this experiment that raise questions.

This point is well taken. As the main focus of the manuscript was on the nanoscale imaging of the hPSC adhesions, we appreciate that the role of the unusual focal adhesions on pluripotency was not carefully explored. We have now studied this point in more detail.

What exactly does culture on these nanogrids alter? For example it is well established through experiments with micropatterning that colony size per se on patterned substrates exercises constraints on self-renewal and differentiation. Do the colonies grow up the edges

of the troughs, or are they confined within them, and if so, how does this work relate to previous studies with micropatterned hPSC culture?

The reviewer is absolutely correct in referring to previous studies where different size micropatterns are used for the hPSC cultures. These studies have shown that restriction of the colony size plays a major role in the differentiation process. However it is hard to extrapolate the results from the micropattern studies directly to nano grating surfaces due to the inherent differences between the systems. These may not have been clearly described and this may have resulted in a misunderstanding: 1) The micropatterned islands are uniform 2D areas coated with adhesive ligands that limit the area the entire colony can occupy. 2) The troughs in the nano grids are very small (800 nm - an average cell has a diameter of 2000 nm and the stem cell colonies shown in figure 1a-b are 4250 μm^2) and hence the colony cells grow on top of the grids. The nano grids limit the area of individual ECM contact sites by separating them physically from each other with 3D grooves. In this system the colony size is not restricted and the cells can expand over the nano grid surface area freely.

Moreover, it is not clear to me that other aspects of subcellular organization might not be disrupted by culture of hPSC on the grid structure. Mechanical forces that impact signal transduction in epithelial cells are mediated not only by interactions with the extracellular matrix through focal adhesions, but also through cell-cell junctions. It is well known that cell-cell contacts are essential for maintenance of pluripotency in hPSC. Importantly it is not clear whether cell-cell junctions are properly maintained on the grid under these conditions, or for that matter, if other parameters apart from focal adhesion size are not altered.

We have investigated this important point with new experimentation. Growing hPSCs on uniform surfaces or grids had no obvious effects on E-cadherin localization to cell-cell junctions (new supplementary Figure 1). Next we investigated the protein levels of E-cadherin in cells grown on uniform surface, nano-grating surfaces or treated with BMP-4 on uniform surface as a positive control for differentiation. In the absence of BMP-4 no difference was detected in 1 or 3 days of culture. Since differentiation marker SSEA-1 is clearly induced in 3 days, this suggests that loss of E-cadherin is not the trigger for differentiation on the nano grids. However, 6 days of culture on the nano grids or 3-6 days of BMP-4 significantly induced E-cadherin levels (Supplementary Figure 1). These data suggest that the nano grids do not dramatically disrupt E-cadherin mediated cell-cell adhesions.

Finally the endpoint is difficult to interpret. It seems likely that culture on the grid (whatever mechanism is invoked to account for its effects) is accelerating a differentiation process that

will eventually occur in control cells as well in the absence of appropriate growth factors. This complex endpoint of accelerated differentiation (the authors studied only one time point which oversimplifies matters) could be modified by many other factors apart from focal adhesion size.

We have performed more careful time course investigation of the effects of the nano grids on differentiation in cells grown on uniform surface, nano-grated surfaces or treated with BMB4 on uniform surface (as a positive control for differentiation) for 1, 3 and 6 days. Our new data (Figure 2) show that 1 day of culture on the nano grids is sufficient to reduce Oct4 levels while 3 days of culture strongly reduced Sox2 levels while increasing SSEA1. These data further support our initial observation that nano grids accelerate differentiation. However, we acknowledge in the text on page 5 that these experiments are not formally proof of a direct link between FA size/architecture and pluripotency “These data suggest that physical constriction of FA size and orientation in hPSC colonies could be sufficient to compromise pluripotency and accelerate differentiation. However, more work would be needed for definitive proof of a direct causal relationship between FA size and orientation and maintenance of pluripotency.”

These new data are in line with a recent publication (that we had regrettably missed earlier but which we are now cite on page 5) using a system comparable to ours (Abagnale et al. Stem Cell Reports 8;9(2):654-666). Here the authors cultured hPSCs on similar scale ECM coated nano grids and find similar effects on pluripotency. hPSCs respond to morphogen BMP-4 faster when cultured on 650nm grooves compared to uniform substrate. They also observe similar colony morphology and reduction in focal adhesion size when cells are cultured on different nanotopographies. The authors also performed gene expression profiling comparing cells grown on uniform surface and 650nm grooves. In pluripotency supporting media (hPSC-Brew, TeaSR-E8) no significant alterations in gene expression were detected. This implies that the nanopatterning alone is not sufficient to alter gene expression or the pluripotent state.

2. In supporting their claims to identify features of the focal adhesions in hPSC that appear to be unique, it is not clear to me whether the authors have carried out appropriate comparisons. hPSC display properties typical of epithelia, but the authors appear to compare them with fibroblasts and with osteosarcoma cells. It would be more informative to carry out these comparisons with a typical diploid cultured epithelial cell such as a keratinocyte. Then it would be more clear if the structure of focal adhesions reported here is really unique to pluripotent stem cells.

We agree with the reviewer that ideally epithelial cells would be a meaningful comparison. However, we would respectfully like to emphasize that acquiring the super resolution data on iPALM is extremely tedious and challenging. Imaging one cell with one of the markers can take up to 6 hours and the data analyses 3-times this. Therefore, our data is only from the hPSC. The comparison to fibroblasts and U2OS cells is based on the fact that “classical” focal adhesions have been previously imaged with the same technique by others (*Nature*. 468:580–584.; *Nat Cell Biol*. 17:880–892) in these cell types.

3. The experiments on kank knockdown are certainly interesting, in that they show a change in the size distribution of focal adhesions within a colony. However, it is curious that given the experiments in Figure 1, that the authors did not further investigate the effects on kank knockdown on cell differentiation. (Such experiments would also have to take into account possible concomitant effects on cell cell adhesion). It would seem that this would be a more direct approach to investigating the role of the large focal adhesions in maintaining pluripotency than the experiments with the nanogrids.

We agree with this point and have now extensively evaluated the effect of Kanks on cell differentiation using silencing of Kank1 and Kank2. We first analysed the effect of Kank1 and Kank2 double silencing on Sox2 using western blot and found no obvious difference in cells cultured in E8 (Figure for reviewer#1, a). Next we investigated cell surface differentiation markers SSEA-1 and pluripotency markers Nanog and Sox2 using FACS in Kank1 and Kank2 double silenced cells grown in E8 (BMP-4 was used as a positive control). No significant differences were induced by KANK-silencing (Figure for reviewer#1 b). Finally we tested long term silencing of Kank1 and Kank2 alone or in duplicate in cells grown in E6. Unfortunately, no obvious changes in Oct4, Nanog, Sox2 were apparent (Figure for reviewer#1 c). Therefore, we conclude that loss of Kanks alone is not sufficient to differentiate, albeit their depletion affects the size distribution of focal adhesions in hPSC colonies. These negative data could be explained by the fact that the effect of KANK-silencing on the hPSC adhesions is not as dramatic as that of the nano grids. In addition, interpretation of these experiments are complicated by the fact that we are unable to specifically silence the long isoform of Kank1. As we show in Supplementary figure 2, hPSC upregulate the long isoform of Kank1 compared to the differentiated cells (in the iPALM experiments the KANK1-L was chosen for imaging based on this notion). This implies that Kank1-L could have a pluripotency specific role and the relative levels of Kank1-L and -S could be biologically important.

Figure for reviewer#1 a) Representative western blot analysis of Kank1, Kank2, Tubulin, Sox2 and GAPDH protein levels in hPSC transfected with either control siRNA (siCTRL) or a combination of siRNA targeting Kank1 and Kank2 (siKank1 + siKank2) cultured in Essential 8 medium for 3 days.

b) Flow cytometry analysis of hPSCs infected with lentivirus; control shRNA (nt control) or combination of shRNAs targeting Kank1 and Kank2 (K1+K2) and cultured for 9 days in Essential 8 medium. Cells treated with BMP-4 are included as a positive control for differentiation. Samples stained with surface marker antibodies TRA-1-60 and SSEA1 (top panel) or with nuclear pluripotency marker Sox2 and Nanog (bottom panel).

c) Western blot analysis of pluripotency factors, Kank1 and Kank2 in hPSCs following 6 days of transfection with control siRNA (siCTRL), a combination of siRNAs targeting Kank1 and Kank2 (siKank1 + siKank2) or individual siRNA targeting either Kank1 (siKank1) or Kank2 (siKank2) cultured in Essential 6 basal medium.

Reviewer #2 (Remarks to the Author):

The paper reports 3 main findings: 1) using nanopatterns to demonstrate that the physical restriction of adhesion size in hPSC colonies is sufficient to trigger differentiation; 2) dissect the nano-structure of these cornerstone adhesions that are gatekeepers for pluripotency; 3) depletion of kanks diminishes the patterning of the cornerstone adhesions. While defining and characterizing cornerstone adhesions and their relationship to pluripotency is novel and significant, there are numerous issues that need to be addressed before this paper could be recommended for publication. Some of the discussion points are not supported by data included in the paper. Either data needs to be added or the discussion needs to be modified. Similarly, throughout the text information is presented that does not seem to connect to the story being presented, but raise questions. Again, these should be addressed or eliminated (eg., location of activated beta1 integrin – why is this the only integrin whose activity was tested)? Finally, as documented below, the colors in the figures, and the discrepancies between Figs 6a and 6b need to be addressed.

We would like to thank the reviewer for these positive, constructive and helpful comments.

Specific points:

It seems that the cornerstone adhesions are structurally and functionally distinct from focal adhesions. Therefore, referring to them as unusually large focal adhesions is confusing, and flipping back and forth between terms makes reading difficult and reduces the impact of the finding. Would recommend using cornerstone adhesion and clearly establishing the definition of this type of adhesion with the data presented in the paper. However, this raises the question of why the particular molecules were investigated with iPALM.

We agree fully with this suggestion and have now modified the manuscript according to this suggestion - using cornerstone adhesions.

A previous paper from this group (Närvä et al, 2017) states that cell-cell junctions (E-Cadherin) contributed to actin fence integrity. Why wasn't E-Cadherin investigated with iPALM? Or, one of the other modes of microscopy used in the paper?

This is a valid point. E-Cadherin was not investigated by iPALM as it detects signal only a few hundred nanometers from the coverslip. As the images in our previous publication indicate, that junctional cadherin is much higher in localization and therefore too far to be imaged correctly with iPALM. This is one of the limitations of this system. A previous study has specifically studied cadherin junctions using iPALM. However, for this set-up the junction had to be generated "artificially" between coverslip tethered E-Cadherin and an individual

MDCK cell (Nat Cell Biol. 2017 Jan; 19(1): 28–37.). Since this setup would disrupt the cornerstone adhesions and would not allow to study the junctions in the colony encircled by an actin fence, we do not think that this system would be useful for investigating the functions of hPSC colonies. However, we have investigated this important point with new experimentation using the AiryScan confocal. In Figure 1 we show that focal adhesion and colony architecture are altered when cells are grown on nano-grated VN-coated surfaces instead of uniform surfaces. However, this does not seem to influence the E-Cadherin junctions. There was no obvious differences in E-cadherin localization to cell-cell junctions (new Supplementary Figure 1) in either condition. We note, however, that the junctions are more elongated on the nano-grated surfaces in line with the more spread cell morphology.

The explanation of the vinculin results is highly speculative. The Case and Xia references use mutants to support their claims. Moreover, the possible interaction of vinculin with alpha-actinin was given as an explanation for the “inverted orientation” of vinculin in the Xia paper, but it is not cited as such. Referencing should be more accurate and without mutant support, claims should be dialed back to match the level of included data.

We agree with this point. We have now referenced the Xia paper more accurately (Page 7 and below). In addition, since we do not know the reason for the inverted vinculin orientation, but can only speculate, we have now toned down the wording and also include the detailed iPALM values linked to the speculation (page 8 and below).

“This very peculiar orientation of vinculin was also recently reported in mouse embryonic stem cells FA in cells plated on laminin, but not on gelatin or fibronectin (Xia et al., 2019). A “head-above-tail” vinculin conformation could be indicative of an extreme activation state, potentially triggered by the high forces exerted in cornerstone FA (Närvä et al., 2017). Indeed, vinculin has been reported to dissociate from talin in cases where forces borne by an individual protein exceed 25 pN (Yao et al., 2014). In such cases, we speculate that vinculin head could bind a different partner that localises higher up in cornerstone FA (such as α -actinin-1). The vertical position of the first α -actinin-1 ($Z_{\text{centre}}=125.5\pm 15.1\text{nm}$) layer partially overlaps with the distribution of the vinculin-N ($Z_{\text{centre}}=100.4\pm 14.4\text{nm}$). Alternatively, the vinculin tail could engages components close to the plasma membrane such as paxillin (Carisey and Ballestrem, 2011). The vertical position of the paxillin ($Z_{\text{centre}}=55.7 \pm 8.5\text{nm}$) partially overlaps with the distribution of the vinculin-C ($Z_{\text{centre}}=76.4 \pm 16.3 \text{ nm}$).”

If the speculation is that vinculin is interacting with alpha-actinin instead of talin, why are data (numbers) not included in the text? This was done with alpha-actinin and actin.

Thank you for this suggestion. As shown in the updated text included above, we have now included the numbers. We also provide all the values from the iPALM data in a

supplementary summary table (Supplementary table 1 and below). There is some partial overlap between the z-distributions of vinculin- N and alpha-actinin and vinculin-C and paxillin.

Protein	Average $Z_{\text{Centre}} \pm$ S.D. (nm)	# localisations	of # of FA	# of colonies
Integrin α V	45.3 \pm 7.6	3.33E+05	38	6
Integrin β 5	46.5 \pm 10.2	5.25E+05	29	5
Paxillin	55.7 \pm 8.5	1.23E+06	64	7
Talin-1-N	72.8 \pm 12.3	1.91E+06	75	11
Talin-1-C	103.5 \pm 20.5	2.42E+05	11	3
Vinculin-N	100.4 \pm 14.4	2.44E+06	71	7
Vinculin-C	76.4 \pm 16.3	6.41E+05	41	5
Actin-Low	99.6 \pm 14.4	3.88E+06	78	11
Actin-High	155.4 \pm 14.2			
α -Actinin-1- Low	125.5 \pm 15.1	1.10E+06	60	6
α -Actinin-1-High	174.8 \pm 17.1			
Kank-1 Adjacent	116.1 \pm 12.1	2.34E+06	33	6
Kank-1 Distal	63.4 \pm 13.6			
Kank-2	100.8 \pm 13.4	2.14E+06	23	3

Adjacent

Kank-2 Distal 74.2 ± 17.7

In the vinculin knockdown does the second layer of actin change?

This is a very interesting point. Given that there is only 2 iPALM microscopes available globally and we do not currently have access to the iPALM, we investigated this using another super-resolution modality, 3D STED. Using 3D STED we could detect peaks in actin intensities that matches the Z coordinates measured using IPALM. However we are very much at the resolution limit of our STED microscope (z resolution of 80 nm) and we are reluctant to make too strong conclusion based on these images alone. This dual actin intensity peaks were not visible / resolved in all our 3D STED images while we could see them in all our IPALM images. This is why these figures were included for the reviewers only. Of Note, on the reviewer figure #2_1 b) the white box highlight the Z volume that can be imaged using IPALM. Regardless we then carried on analysing the effect of vinculin silencing as suggested. Vinculin shRNA HEL23.4 cells were compared with control shRNA infected HEL23.4 and analysed with STED (Figure for reviewer#2_2). Vinculin silenced cells also displayed a visible actin double layer.

Figure for reviewer #2 1. a) Representative confocal microscopy image of an individual actin stress fibre in hPSC colony (HEL24.3). F-actin (magenta) stained with Sir-Actin and the focal adhesions (green) with paxillin antibody. The arrows represent orientation and scale of the individual panels.

b) 3D STED microscopy image of the same stress fibre shown in (a). The arrows represent orientation and scale the of the image. White box marks the area used for

the sum intensity profile. The white box also represent the typical z volume that can be imaged using IPALM. c) Sum intensity profile of the F-actin (magenta) and Paxillin (green) staining. The area (white box) and direction (yellow arrow) of the measurement can be observed from the panel (b).

d) Representative confocal microscopy image of an individual focal adhesion in hPSC colony (HEL11.4). F-actin stained with Sir-Actin and the focal adhesions with paxillin antibody. The arrows represent orientation and scale of the individual panels. e) 3D STED microscopy image of the same focal adhesion shown in (d). The arrows represent the orientation of the image. White box marks the area used for the sum intensity profile. f) Sum intensity profile of the F-actin and Paxillin staining. The area (white box) and direction of the measurement (yellow arrow) can be observed from the panel (e).

Figure for reviewer #2 2. a) Representative confocal and 3D Sted microscopy images of an individual focal adhesion in hPSC colony expressing control shRNA (shCTRL). F-actin (magenta) stained with Sir-Actin and the focal adhesions (green) with paxillin antibody. The arrows represent orientation and scale of the individual panels. b) Sum intensity profile of the F-actin staining. The area (white box) and direction of the measurement (yellow arrow) can be observed from the panel (a). c) Representative confocal and 3D STED microscopy images of an individual focal adhesion in hPSC colony expressing Vinculin targeting shRNA (shVCL55). F-actin (magenta) stained with Sir-Actin and the focal adhesions (green) with paxillin antibody. The arrows represent orientation and scale of the individual panels. d) Sum intensity profile of the F-actin staining. The area (white box) and direction of the measurement (yellow arrow) can be observed from the panel (c).

If kinks regulate microtubule targeting, what happens to microtubules? Is there interplay between microtubules and the actin cables? If it is outside the scope, then clarify.

We agree fully that this would be an interesting avenue of investigation. However, given to focus of the paper - the nanoscale architecture of the cornerstone focal adhesions - and the fact that iPALM is not the instrument to image microtubules (due to their distance from the coverslip), we consider this to be outside of the scope of this manuscript.

Cornerstone adhesion dynamics are not quantified. They are presented in a single movie, where the adhesions on the right seem much slower to turnover than the adhesions on the left. Quantification is needed for multiple experiments.

We agree that this is an important point. In new figure 1 c-g we present careful quantitative analyses of the focal adhesion dynamics from 3 experiments and 16 colonies. We find that a significantly higher subpopulation of the edge focal adhesion have an extremely long life time (over 105 minutes) compared to the center focal adhesions (Figure 1f). In addition, large focal adhesions are more frequent at the colony edge compared to the center.

The discussion claims that the size and orientation of cornerstone adhesions are essential for pluripotency. This is not directly shown. While supplemental figures demonstrate that different perturbations that affect pluripotency decrease the size of cornerstone adhesions, or the density around the periphery, a direct causal relationship between them is not established. Therefore, a more careful wording is required.[CG4]

We have performed more careful time course investigation of the effects of the nano grids (that limit focal adhesion size and orientation) on differentiation in cells grown on uniform surface, nano-grated surfaces or treated with BMP-4 on uniform surface (as a positive control for differentiation) for 1, 3 and 6 days. Our new data (Figure 2) show that 1 day of culture on the nano-grids is sufficient to reduce Oct4 levels while 3 days of culture drastically reduced Sox2 levels while increasing SSEA1. These data further supports our initial observation that altering focal adhesion size/orientation accelerate spontaneous differentiation. However, we acknowledge the limitation of these data and mention on page 5 that these are data are not definitive proof of a direct causal relationship between focal adhesion size and orientation and maintenance of pluripotency. "These data suggest that physical constriction of FA size and orientation in hPSC colonies could alone be sufficient to compromise pluripotency and accelerate differentiation. However, more work would be needed for definitive proof of a direct causal relationship between FA size and orientation and maintenance of pluripotency. "

No details are given about the reconstruction of single molecule images. What were the lateral and axial localization precisions? The very large clusters in Fig 2a suggest different localization precision than Fig 2b, and Fig 2c seems (by eye) different yet again.

The typical lateral localization precision (X and Y) is about ± 20 nm as this only depends on how well we can determine the central position of all the localizations (blink). As for the axial (Z) localization precision, this one varies depending on how close we are to the glass (central focus) going from ± 10 nm up to ± 20 nm or more. All the information on axial localization is in the supplementary table.

As for the differences in the size of clusters between figure 2a, 2b and 2c (currently Figures 3a-c), this is not due to the localization precision but to the density of localizations we had in the particular area. All these images are renderings of the localizations that were measured (not the localizations themselves) and the size of the clusters only reflects the density of localizations (number of localizations). In figure 2a (currently 3a) the fluorescent protein expression was low and we have areas with many localizations and others with almost none, while in figure 2b (currently figure 3b), the expression is much better and localizations are well distributed. Please note that figure 2a (currently figure 3a) is a slightly different magnification, as indicated by the size of the scale bars.

Some of the data, vinculin and kanks, are claimed to have axial distributions that vary laterally. This is nicely demonstrated for vinculin in Fig 3e, one wonders why something similar was not done the kanks?

We would respectfully highlight that Figure 5 (currently figure 6) shows how the axial distributions of kanks vary in the vicinity of the focal adhesions and further away from the adhesions (albeit the presentation of the data is different to the one for vinculin in Figure 4). Furthermore, the supplementary video 2 shows the distribution in 3D clearly.

Is all of beta1 active? Why does it not localize with talin, one of its activators? Is the activity of other integrins constrained?

We find that total $\beta 1$ -integrin is diffusely localised throughout the adhesion (New Supplementary Figure 7), similarly to the active $\beta 1$ -integrin. Hence, the localisation of the active seems to follow to distribution of the overall $\beta 1$ -pool. Indeed, talin is more clearly localized to the rims of the focal adhesion similarly to $\alpha v\beta 5$ -integrin. However, there is some talin localized throughout the adhesions, suggesting that talin would be available throughout the adhesion to activate integrins (Figure 3c, Supplementary Figure 5c, d). Unfortunately, we

do not have access to activation specific antibodies available for imaging α v-integrins, limiting our ability to analyse their activity in detail.

Why wasn't the other VN receptor alpha v beta3 investigated? The paper mentions using beta 5 and alpha v beta 5 antibodies, why not alpha v beta 3 and alpha v?

Initially, we had investigated only α v β 5-integrin. The reason for this was that earlier work investigating hPSC adhesion to vitronectin has demonstrated a key role for this vitronectin binding integrin (Braam et al. 2008). We have now investigated the other vitronectin binding integrins, as suggested by the reviewer, by staining endogenous α v-integrin, β 3-integrin and α v β 5-integrin (New Supplementary figure 6 and 8). Interestingly, we do not detect β 3-integrin in the iPSC cornerstone adhesions (New Supplementary 8) even though the same staining protocol detects strong β 3-integrin signal in diploid normal epithelial ARPE-19 cells (New Supplementary 8), indicating that the antibody works. α v-integrin alone is more homogeneously distributed than the α v β 5 integrin antibody and may pair with β 1-integrin in the center of the adhesions (New supplementary figure 6).

In 5e is it focal adhesion or cornerstone adhesion area?

Sorry for not being clear. The focal adhesions at the colony edge are predominantly "cornerstone adhesions" but also smaller adhesions and the focal adhesion in the middle are smaller and more similar to "classical" focal adhesions. As both were analysed here, we chose to label the y-axis in Figure 5f (we assume this is the figure the reviewer was referring to) FA area.

The term horizontal segregation seems inaccurate since there is no fixed coordinate system—spatial segregation to the periphery of adhesion would perhaps be clearer.

We have now used the term spatial segregation, as suggested by the reviewer.

N analyzed and statistics used seem to be missing for the most part.

We agree this is an important point. We have provided all these details for the iPALM data in the supplementary table. As for other quantifications, N numbers and statistics explanations are included in the methods section or described in the figure legends.

There are numerous issues with the figures that need to be addressed.

- Why was Fig 2e repeated in next 3 figures? It took multiple tries to understand that the opacity change was to indicate which layer was being addressed in the rest of the figure.

We apologize for this not being self explanatory. The figure was repeated to help the reader keep track of the layer that we are describing in each figure. This is now indicated more clearly in the figure legends indicating: *“Color bars highlight the z range for each of the three layers. To highlight that data from the X layer is presented here this layer is made full intensity while other layers are displayed as transparencies.”*

- The left-to-right shading of ALL of the box and whiskers does not seem to have any relevance to axial data. Why is it there? There is no explanation for it. Similarly, why are the bars in the repeating Fig 2e shaded left-to-right?

We acknowledge this valid point of the reviewer and would like to explain that this shading is generated by PeakSelector, the software used to determine all the localizations and to create the figures. The function of those tones of shading was to represent the density of localisations in a similar way as the cluster size in the renderings. Here inside the boxes it works as a density scale bar to show the densities going from low (lighter tones) to high (darker tones). In the newest version of the figures all the shading has been removed.

- Why do the axial height color bars in ALL of the figures have left-to-right variations? It is especially obvious in the yellow region around 50 nm.

This question is highly related to the question from the same reviewer presented just before. The left-to-right variations or changes in tonality refer to the density of localisations that the PeakSelector software has found in a particular region of the image. The variations/shading in boxes, whiskers and even in cartoons is present there as a residual of the fact that all color bars were copied from the scale bars of real data with the intention to preserve the same color scheme and maintain the quality of the data. In the newest version of the figures all the shading has been removed.

- What are the boxes under the side views of adhesions in Figs 2a-c? Normalized intensity?

Yes, they are normalized intensity profiles of the side view images. This has now been mentioned in the figure legend (in the current version Figure 3a-c): *“In addition, normalized line intensity profiles of the region of interest, in both z- and x-axis, are shown in red lines.”*

- The right-hand panel of Fig 3d would be more informative as a Vinculin-N. It would match the panel below it, and avoid the confusing changes in color for paxillin and vinculin in Fig 3d.

We would like to clarify that Figure 3d (now Figure 4d) is a two-color iPALM image and that the right panel represents an overlay of the two images/colors (shown in the left and middle

panel) to demonstrate their relative distributions in the same FA. In the panels in the left and the center, the color represents the height according to the scale bar, and in the panel of the right, the color is homogeneous for each protein (red for paxillin and green for Vinculin-C). In the main figure, figure 4e right shows the distribution and Figure 4c the quantification of Vinculin-N.

- Again, why does the color change between Fig 4b left and 4b right?

Please see the response above. In the overlay case we cannot display the proteins in color as a function of the height, but we need to use homogeneous colors for each protein. The default colors of the PeakSelector software are green and red.

- Why don't the layers in Fig 6b match the data in Fig 6a? For example, the first layer of actin is below 100 nm in Fig 6a and above 100 nm in Fig 6b.

We are grateful to the reviewer for pointing out this inaccuracy. The summary figure (old 6b; current 7b) has been updated to match better the measured values shown in 7a (old 6a).

- Why weren't the elevations of proteins in the periphery shown in the cartoon?

We chose to omit these from the cartoon, since the proteins in the periphery are not an integral part of the FA but they are FA-associated proteins and therefore we do not include them in the cartoon.

- Finally, the colors in Fig 6 do not readily distinguish the layers.

We had chosen these colours to show the localization height of each protein, something that is a continuum going from yellow to purple and passing by all the other colors. In order to distinguish each layer we have included the bars on the top of figure 6a that clearly label each layer and the heights that are covered in each layer. It is important to note that the layers are not completely exclusive but that there is a certain level of overlap between them, something that is clearly seen in figures 2e, 3a and 4a and more specifically by looking at the position of the first actin layer that clearly overlaps with Vinculin and Talin-1-C.

Reviewer #3 (Remarks to the Author):

. The authors have used super-resolution microscopy to examine the edges of hESC colonies that have unusual actin 'fences'. In the abstract and introduction, they indicate that the hypothesis is that the atypical Focal adhesions seen at the borders of hESC colonies are gatekeepers to pluripotency. If this is the big question that they sort to address in the paper, then the data do not answer it. They nicely characterise the 'cornerstone' adhesions by super resolution microscopy techniques and show some similarities/differences to some previous reports but it's not clear what that means for these cornerstone adhesions in terms of function – here in relation to pluripotency / fate determination signals. They do add in a new part in that no-one has done this sort of super-resolution microscopy for Kank before.

Overall, the study seems to have been done competently although I am not a super-resolution expert, but the major question is unclear and the hypothesis that these cornerstone adhesions are responsible for maintaining pluripotency is not really addressed. We are grateful for the reviewer for these constructive comments. We would like to respectfully point out that the main point/major question of this paper is the detailed nanoscale architecture of the corner stone adhesions. We agree that it is important to determine in the detail how these adhesion contribute to pluripotency. However, this is not the main point of our manuscript and is somewhat beyond the scope of this study.

The paper seems to be reporting a number of interesting observations but neither linking them all or focussing on one question. How relevant their findings are since they use only 1 hESC line, perform limited nano-surface experiments and then do not link the 2 parts of the paper together is a question.

We would like to point out that in our original manuscript describing the observation of the cornerstone adhesions and the actin ring (Närvä et al., 2017), we validated the generality of this observation in two different iPSC lines and well as hESCs. However, since we agree that this is an important point, we have validated the following key observations in two different iPSC lines: 1) Talin localising predominantly to the focal adhesion edges (New Supplementary Figure 5); 2) β 1-integrin localising more to the center of the adhesion compared to β 5-integrin (New Supplementary Figure 7); 3) The actin double layer (using 3D STED, see our response to reviewer#2 and below).

Figure for reviewer. a) Representative confocal microscopy image of an individual actin stress fibre in hPSC colony (HEL24.3). F-actin (magenta) stained with Sir-Actin and the focal adhesions (green) with paxillin antibody. The arrows represent orientation and scale of the individual panels.

b) 3D STED microscopy image of the same stress fibre shown in (a). The arrows represent orientation and scale the of the image. White box marks the area used for

the sum intensity profile. c) Sum intensity profile of the F-actin (magenta) and Paxillin (green) staining. The area (white box) and direction (yellow arrow) of the measurement can be observed from the panel (b).

d) Representative confocal microscopy image of an individual focal adhesioin in hPSC colony (HEL11.4). F-actin stained with Sir-Actin and the focal adhesions with paxillin antibody. The arrows represent orientation and scale of the individual panels.

e) 3D STED microscopy image of the same focal adhesion shown in (d). The arrows represent the orientation of the image. White box marks the area used for the sum intensity profile. f) Sum intensity profile of the F-actin and Paxillin staining. The area (white box) and direction of the measurement (yellow arrow) can be observed from the panel (e).

In addition, we have studied the effects of the nano grids on differentiation using two iPSC lines and find that they behave similarly. These data are included below for the reviewer and in New Figure 2E.

We agree with the reviewer that ideally all the iPALM measurements could have been performed in two cell lines. However, we would respectfully like to emphasize that acquiring the super resolution data on iPALM is extremely tedious and challenging. Imaging one cell with one of the markers can take up to 6 hours and the data analyses 3-times this. Therefore, our iPALM data is, regrettably, only from a single iPSC line.

1. They are a reputable group with experienced imaging specialists and seem to be doing things the right way. I do worry, however, that they may be pushing the limits of the data

interpretation. For example in figure 3 panel e they describe concave ('cup-like') adhesion structures. How can they be sure this is not an imaging artefact?

We appreciate the compliments from this reviewer and we acknowledge that we certainly are close to the limits of interpretations that the data allows, but still within the boundaries. We are certain that the concave distribution is not due to imaging artefacts as it is only present in Vinculin and not in any of the other proteins we studied (please see figure 4e (old 3e) of the manuscript for a control comparison with Paxillin). Furthermore the described "cup-like" distribution was seen in Vinculin tagged at different ends of the protein (Vinculin-N and Vinculin-C in figure 4e) giving more support to the idea that this is real and not an imaging artefact. Also we would like to emphasise that the same distribution is observed in multiple focal adhesions and with millions of localisations in each case giving us the certitude that the distribution is indeed concave and not flat as in other proteins.

2. Figure 1 describes data to suggest that nano-grated surfaces alter the organisation of cornerstone adhesions resulting in altered cell fate responses. They don't have any evidence for this. How can they be sure that it is the altered cornerstone adhesions that result in the effects observed in panels d-f on Sox2 and SSEA-1 expression? Many alternative explanations for these effects are surely possible. For example, the pluripotent colonies comprise many cells with numerous cell-cell interactions. Are there changes in cell-cell adhesions that cause these effects? Cell-cell adhesion changes are well known to drastically affect hESCs,

We have investigated this important point with new experimentation. Growing hPSC on uniform surfaces or grids had no obvious effects on E-cadherin localization to cell-cell junctions (new Supplementary Figure 1). We also investigated the protein levels of E-cadherin in cells grown on uniform surface, nano-grated surfaces or treated with BMB4 on uniform surface as a positive control for differentiation. In the absence of BMP-4 no difference was detected in 1 or 3 days of culture. However, 6 days of culture on the nano grids or 3-6 days of BMB-4 significantly induced E-cadherin levels (New supplementary Figure 1). These data suggest that the nano grids do not dramatically disrupt E-cadherin mediated cell-cell adhesions. We have also performed more careful time course investigation of the effects of the nano grids on differentiation in cells grown on uniform surface, nano-grated surfaces or treated with BMB4 on uniform surface (as a positive control for differentiation) for 1, 3 and 6 days. Our new data (Figure 2) show that 1 day of culture on the nano grids is sufficient to reduce Oct4 levels while 3 days of culture drastically reduced Sox2 levels while increasing SSE1. These data further supports our initial observation that nano grids accelerate differentiation. However, the nano grid induced differentiation is slower than BMP-4 induced.

These new data are in line with a recent publication (that we had regrettably missed earlier but which we are now cite on page 5 and 10) using a system comparable to ours (Abagnale et al. Stem Cell Reports 8;9(2):654-666). Here the authors cultured hPSCs on similar scale ECM coated nano grids and find similar effects on pluripotency. hPSCs respond to morphogen BMP-4 faster when cultured on 650nm grooves compared to uniform substrate. They also observe similar colony morphology and reduction in focal adhesion size when cells are cultured on different nanotopographies.

3. Furthermore, loss of two markers when cells were transferred into E6 (differentiation base medium without stem cell maintaining activin and FGF) cannot be constituted as loss of pluripotency and it is notable that Oct 4 and Nanog are not examined. Do these cells differentiate to progenitors quicker than with BMP?

Please see the response above.

Importantly the authors did not look at transferring the colonies back into stem cell maintenance medium from E6...do they recover these 2 markers just as quickly. In mESC it has previously been shown by Chambers and colleagues that even Nanog can be lost for days without changing pluripotent status on recovery.

We tested this with new experimentation and find that the cells do not recover these markers when returned to stem cell maintenance medium. Please see new Supplementary Figure 1 d-f.

4. Figure 1 only describes one type of nano-grated surface. Why were these dimensions chosen? What happens when they are altered? A more detailed analysis of these parameters together with the resultant quantification of the cornerstone adhesions would benefit the study. Also how is the nanoscale architecture of the cornerstone adhesions altered by the nano-grated surface? Why isn't this linked and compared to the pluripotency marker results?

The dimension of the nano-grated surfaces were chosen for two reasons: 1) the size is ideal to restrict the formation of the largest FA and therefore only the formation of cornerstone adhesion is disrupted. 2) these patterns are commercially available on high quality coverslip which allow the imaging of our cells at high resolution using airyscan microscopes. Regrettably, these nano-grated pattern are not compatible with IPALM imaging (or other super-resolution modality such as SIM or STED). IPALM in particular detects signal only a few hundred nanometers from the lowest point of the coverslip and works only with very specialized gold-particle embedded coverslips. Unfortunately, this is one of the limitations of this system.

5. The whole paper seems to be performed with only one hESC stem cell line. This is not the standard for work with human pluripotent stem cells (with very different genetic background). How representative are these findings to other stem cell lines? Key experiments at minimum should be validated in a second line. Please see our response above.

6. IF panels in figure 2 and onwards appear to be of individual cornerstone adhesions but this is not made clear in the manuscript. It should be clarified for the reader so that they can easily relate these images to the part of hESC colonies that is being shown.

We agree and this is now clearly indicated in the figure legends and in the text.

7. Nick Brown in Cambridge has shown different orientations of talin by super resolution microscopy in flies. This paper could also be referenced and the findings discussed. Klapholz et al Current Biology Volume 25, <https://www.sciencedirect.com/science/article/pii/S0960982215000755#sec2>

We thank the reviewer for pointing out this work and we have now included it in the introduction on page 3.

8. Vitillo and colleagues Stem Cell Reports 2016 also published data implicating FAK in pluripotent stem cell maintenance using a number of lines. This work should also be referenced

[https://www.cell.com/stem-cell-reports/fulltext/S2213-6711\(16\)30130-8](https://www.cell.com/stem-cell-reports/fulltext/S2213-6711(16)30130-8)

We thank the reviewer for pointing out this work and we have now included it in the introduction on page 3.

Reviewers' comments:

Reviewer #1 (Remarks to the Author):

In their revised manuscript and rebuttal, the authors have provided thoughtful and thorough responses to most of the major concerns raised in the first round of review. There are a couple of issues that I think still require further clarification.

1. Thanks for the explanation of the structure of the nanogrids and the interaction of the cells with this substrate. However, if I have followed the results correctly, the nanogrid cultures do seem to bend the cell and colony orientation and shape somewhat. It is known that enforced alterations to cell orientation and colony shape at the borders can influence cell behavior. I would just suggest some caution in interpreting these studies; the perturbations caused by nanogrid culture may well affect other structural parameters independently of action on the adhesions.

2. Although I appreciate the difficulty in carrying out the IPALM technique, it would presumably not be too difficult to examine an epithelial cell using the techniques employed in Figure 1, which rely on routine confocal microscopy and structured illumination. The point is important because it would be very interesting to know to what extent these structures were really a unique feature of pluripotent cell.

Reviewer #2 (Remarks to the Author):

The authors have satisfactorily addressed all of the questions raised in the previous round of review, and I am delighted to recommend publication.

Reviewer #3 (Remarks to the Author):

The authors seem to have toned down their claims about the nanotopology regulating pluripotency to some extent, although what 'guarding pluripotency' (introduction) is not at all clear and should be rephrased to 'conducive to' or similar. To this reviewer the new Fig 2 indicates that there is a substantial reduction in Oct 4 and Nanog on the uniform surface which is almost comparable to that on the nanograded surface so that the nano surface affect is pretty limited. They really cannot say that it has any major effect on differentiation. The authors have also reported at least some of the talin and integrin localisation data on 2 further lines and these localisation experiments are convincing. I'm not convinced that given enough data they would not have seen change in E-cadherin (Suppl Fig 1). The imaging is only on d1 and Westerns will look at cells from whole colonies, while there affects are supposed to relate only to edges/corner focal adhesions so will likely be invisible by these techniques. There is interesting data on the architecture and localisation of components in the cornerstone focal adhesions which warrants publication but what it means in terms of cell behaviour and lineage commitment is completely uncertain.

Point-by-point response to reviewer's comments

Reviewers' comments:

Reviewer #1 (Remarks to the Author):

In their revised manuscript and rebuttal, the authors have provided thoughtful and thorough responses to most of the major concerns raised in the first round of review. There are a couple of issues that I think still require further clarification.

We thank the reviewer for acknowledging our efforts in experimentally addressing all the points raised by the reviewers. Please find below our responses to the remaining issues.

1. Thanks for the explanation of the structure of the nanogrids and the interaction of the cells with this substrate. However, if I have followed the results correctly, the nanogrid cultures do seem to bend the cell and colony orientation and shape somewhat. It is known that enforced alterations to cell orientation and colony shape at the borders can influence cell behavior. I would just suggest some caution in interpreting these studies; the perturbations caused by nanogrid culture may well affect other structural parameters independently of action on the adhesions.

We agree that the somewhat altered, less round, colony shape on the nano-grids may have impact on the cells independently of the adhesions. To acknowledge this we have included the following sentence on page 5 "However, it remains possible that the nano-grids affect other structural parameters than the cornerstone FA and these may also contribute to the observed cellular responses".

2. Although I appreciate the difficulty in carrying out the IPALM technique, it would presumably not be too difficult to examine an epithelial cell using the techniques employed in Figure 1, which rely on routine confocal microscopy and structured illumination. The point is important because it would be very interesting to know to what extent these structures were really a unique feature of pluripotent cell.

We thank the reviewer for clarifying that imaging the epithelial cells with microscope modalities other than iPALM would be of interest. This is certainly feasible and we agree that comparison between the hPSC and normal epithelial cells is interesting. We have now imaged ARPE-19 cells using Spinning disk confocal to further validate that the structures of interest are somewhat unique to hPSCs. Supplementary figure 1 now contains new panel a showcasing the actin and focal adhesion morphology in ARPE-19 cells. In addition, we have added the following sentence on page 5 "Importantly cornerstone FA or actin fences are not a general feature of colony forming cells as they were not observed in normal epithelial cells (Supplementary Fig 1a)".

Supplementary figures 5 and 6 both have additional panel e of Spinning disk images of ARPE-19 cells stained either with Talin antibody (Supplementary 5e) or Integrin antibodies (Supplementary 6e) emphasizing the fact that the ring of talin in periphery of FAs and spatial segregation of integrin subunits are not detectable in ARPE-19 cells. We have inserted following sentences on page 7 to address these observations "Interestingly the ring-like distribution of talin could not be observed in normal epithelial cells (Supplementary Fig. 5e and 5f) and $\alpha V\beta 5$ integrin and $\beta 1$ integrin predominantly segregated to distinct FA rather than being segregated into different subdomains

within the same FA (Supplementary Fig 6e). Taken together, in hPSC cornerstone FA, the integrin signalling layer appears to be horizontally segregated into sub-regions of different VTN-binding integrins.”

Reviewer #2 (Remarks to the Author):

The authors have satisfactorily addressed all of the questions raised in the previous round of review, and I am delighted to recommend publication.

We thank the reviewer for these enthusiastic and supportive comments and for finding our super-resolution imaging interesting and worthy of publication.

Reviewer #3 (Remarks to the Author):

The authors seem to have toned down their claims about the nanotopology regulating pluripotency to some extent, although what ‘guarding pluripotency’ (introduction) is not at all clear and should be rephrased to ‘conducive to’ or similar.

We fully agree with the reviewer that we should further tone down the claims and remove the term “guarding pluripotency”. We have changed the title and the new title is **“Superresolution architecture of cornerstone focal adhesions in human pluripotent stem cells”**. We have also changed the abstract by removing the previous nano-grid related claims and shifting the emphasis to the cornerstone imaging and the unique adhesion features unveiled by it.

To this reviewer the new Fig 2 indicates that there is a substantial reduction in Oct 4 and Nanog on the uniform surface which is almost comparable to that on the nanograded surface so that the nano surface affect is pretty limited. They really cannot say that it has any major effect on differentiation.

We agree that the effect is small, albeit significant for Sox2 and Oct4. We also observe clear trend of more rapid reduction of Nanog and clear increase of differentiation marker SSEA-1. We have further toned down our conclusions. On page 5 we write “exposure to VTN-coated nano-grids accelerated, to some extent, spontaneous differentiation as observed by a decrease in the pluripotency factor Oct4 (day 1), Sox2 (day 3 and day 6) and, conversely, an increase in the level of the differentiation marker SSEA-1.”

The authors have also reported at least some of the talin and integrin localisation data on 2 further lines and these localisation experiments are convincing.

We are happy, that these new imaging data were convincing.

I’m not convinced that given enough data they would not have seen change in E-cadherin (Suppl Fig 1). The imaging is only on d1 and Westerns will look at cells from whole colonies, while there affects are supposed to relate only to edges/corner focal adhesions so will likely be invisible by these techniques.

We have included an additional image focusing on the E-cadherin staining on the colony edge (Supplementary figure 1d) and based on this the cell-cell adhesion also at the colony edge seems

largely intact. We agree that it is possible that the junctions become altered over time which is known to happen during differentiation. However, if junctional disruption would be the trigger for the faster spontaneous differentiation on the nano-grids, one would expect to see differences already on day 1 when the Sox2 levels are decreasing. Nevertheless, we have modified the text to acknowledge this concern. On page 5 we write “Plating hPSC on nano-grids did not decrease E-cadherin protein levels or affect their ability to form E-cadherin junctions or (Supplementary Fig. 1b, 1c and 1d), suggesting that cell-cell adhesions are not obviously disrupted. However, it remains possible that the nano-grids affect other structural parameters than the cornerstone FA and these may also contribute to the observed cellular responses.”

There is interesting data on the architecture and localisation of components in the cornerstone focal adhesions which warrants publication but what it means in terms of cell behaviour and lineage commitment is completely uncertain.

Thank you for these encouraging comments. We are happy that the reviewer appreciates the value of the imaging and finds our observations on the unique architecture of the cornerstone adhesion interesting. We agree that the link between adhesion architecture and pluripotency remains to be investigated further. Therefore, we have removed all speculation about the link between adhesion and pluripotency maintenance from the discussion (and elsewhere in the article as described above). Instead, we have emphasized further the unique and unexpected nanoscale architectural features of the cornerstone adhesions and discussed them.

REVIEWERS' COMMENTS:

Reviewer #1 (Remarks to the Author):

I am satisfied with the authors' responses to the outstanding queries